
# Bosonic entanglement renormalization circuits from wavelet theory

**Freek Witteveen[1*] and Michael Walter[1,2]**

**1** Korteweg-de Vries Institute for Mathematics and QuSoft,
University of Amsterdam, Netherlands
**2** Institute for Theoretical Physics and Institute for Language, Logic, and Computation,
University of Amsterdam, Netherlands

* f.g.witteveen@uva.nl

## Abstract

Entanglement renormalization is a unitary real-space renormalization scheme. The corresponding quantum circuits or tensor networks are known as MERA, and they are particularly well-suited to describing quantum systems at criticality. In this work we show how to construct Gaussian bosonic quantum circuits that implement entanglement renormalization for ground states of arbitrary free bosonic chains. The construction is based on wavelet theory, and the dispersion relation of the Hamiltonian is translated into a filter design problem. We give a general algorithm that approximately solves this design problem and provide an approximation theory that relates the properties of the filters to the accuracy of the corresponding quantum circuits. Finally, we explain how the continuum limit (a free bosonic quantum field) emerges naturally from the wavelet construction.

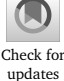

# 1  Introduction

An important task in the study of quantum many-body systems is finding useful parameterizations of physically relevant quantum states. One successful approach is to consider so-called *tensor network states*, which are defined by contractions of local tensors according to a network or graph structure. This gives a natural way to prescribe the entanglement structure of the state, while retaining the ability to describe interesting states such as low energy states of local Hamiltonians. See [1–4] for reviews of tensor network states. Tensor networks are particularly useful to implement real-space renormalization methods for strongly interacting quantum many-body systems. In one spatial dimension, prominent examples are the density matrix renormalization group [5], with the associated tensor network class of matrix product states (MPS) [6] and *entanglement renormalization* [7], with the corresponding multiscale entanglement renormalization ansatz (MERA) states [7,8]. Entanglement renormalization implements a real-space renormalization by a local unitary transformation, decomposing a state into a product state and the renormalized state. By applying many such layers one can build a highly entangled state from product states. Scale-invariant MERA states are a good variational class for approximating ground states of critical quantum chains and one can extract the conformal data of the continuum limit conformal field theory of the system from the entanglement renormalization superoperator [9]. If the entanglement renormalization unitaries are implemented by low-depth local quantum circuits we will call this an *entanglement renormalization circuit* – see Fig. 1 for an illustration. This class of states can be prepared efficiently on a quantum computer, which makes them a promising ansatz class for variational optimization on a quantum computer. This latter perspective was introduced in [10], where the corresponding class was called *DMERA*. The contraction cost using known classical contraction algorithms of such DMERA states increases exponentially with the depth of the quantum circuit, compared to which the contraction of these states is exponentially faster on a quantum computer. Another appealing property of entanglement renormalization circuits is that they are robust to small errors, which makes them interesting candidates for noisy intermediate-scale quantum (NISQ) devices [10,11]. Entanglement renormalization circuits are appealing as their depth is logarithmic in the system size, and the circuit depth of a single layer typically scales polylogarithmically in the desired error, and they apply to gapless systems. See [12,13] for some other applications of tensor networks for quantum computing.

Unfortunately our analytic understanding of MERA in general and DMERA in particular is still limited (as compared to for instance MPS). One direction in which progress to analytic understanding has been made is in connection to wavelets. Wavelet transforms decompose

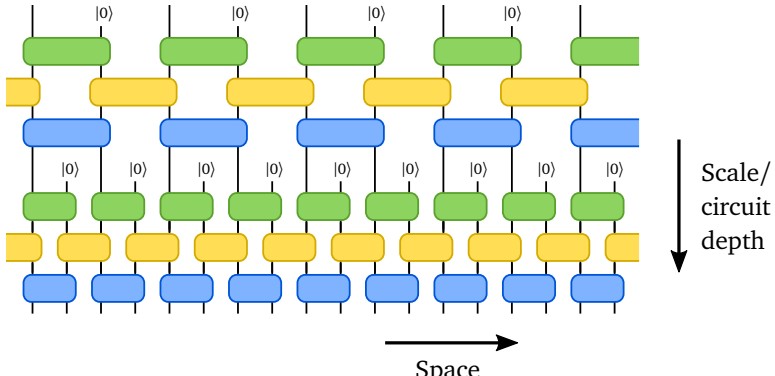

Figure 1: The structure of an entanglement renormalization circuit. Each layer is a constant depth quantum circuit that is supposed to implement a real-space renormalization. Every layer takes as input the output of the previous layer and a product state, resulting in an entangled quantum state at the bottom. Layers further up in the figure correspond to structure at larger scales.

a signal as a linear combination of localized wave packets or 'wavelets' at different scales (as compared to the Fourier transform, which uses plane waves). This can be implemented iteratively: In each step the signal is decomposed into a high-frequency component (the 'details' of the signal) and a low-frequency component (the 'large scale structure' of the signal). The wavelet transform then proceeds iteratively on the low-frequency component of the signal. Wavelet theory has many and wide-ranging applications, from practical signal processing applications such as image compression [14] to mathematical analysis [15]. The procedure of the wavelet transform is very similar to real-space renormalization, and its original development was partially motivated by applications in real-space renormalization.

Recently it has been observed [16–18] that any finite wavelet transform can be written as a classical linear circuit whose fermionic second quantization gives rise to a free fermionic (D)MERA. Moreover, the continuum limit can be precisely related to the corresponding wavelet functions [19]. In [16] it was suggested that a similar result could also be true for free bosonic systems. In order to formulate what this entails, we will work with bosonic quantum circuits. This means that we have a set of bosonic modes, and a bosonic quantum circuit will be a sequence of operations acting locally on these bosonic modes. We restrict to the subclass of Gaussian or linear optics circuits, meaning that each local operation is implemented by time evolution with a quadratic Hamiltonian. This is an efficiently simulable subclass of all bosonic quantum circuits (upon adding non-Gaussian bosonic quantum gates, however, bosonic quantum circuits are able of universal quantum computation [20]). In contrast to more usual notions of quantum circuits and tensor networks, the Hilbert spaces are infinite dimensional. In particular, the usual definition of a tensor network with a finite bond dimension has no immediate analogue. However, finite-depth quantum circuits such as entanglement renormalization circuits of the form of Fig. 1 are still meaningful even in this infinite-dimensional bosonic setup. The notion of Gaussian bosonic entanglement renormalization has been introduced and studied in [21], in which an extensive explanation of the formalism can be found.

## 1.1 Main results

In this work we show that one can indeed construct a Gaussian bosonic entanglement renormalization scheme for bosonic quadratic one-dimensional Hamiltonians, using the second quantization of *biorthogonal wavelet filters* or perfect reconstruction filters. This extends the wavelet-MERA correspondence substantially. The resulting entanglement renormalization takes the form of a short-depth Gaussian bosonic circuit, providing evidence for the relevance of entan-

glement renormalization circuits for preparing ground states of (near) critical quantum systems. Moreover we can relate, similar as in the fermionic case [17], properties of the biorthogonal wavelet transform to the resulting MERA state, and we prove a rigorous approximation theorem for the correlation functions of the MERA state. Interestingly, our formalism is not restricted to the scale-invariant case, but can be used to construct entanglement renormalization circuits for arbitrary translation invariant quadratic bosonic Hamiltonians. Given such a Hamiltonian, we explain how a corresponding (approximate) entanglement renormalization circuit can be found by solving a filter design problem. We also give a general method for constructing such filters, similar to the construction of the Daubechies wavelets. This is in contrast to the fermionic case, where the only known constructions are for massless (critical) fermions [16, 17]. Finally, the continuum limit of the discrete system is directly related to the biorthogonal scaling and wavelet functions corresponding to the filters. For the free massless boson our construction reproduces various scaling dimensions *exactly*. If the system is not scale-invariant, we explain how one can still define versions of the wavelet and scaling functions which are not scale-invariant.

A natural application of a quantum computer based on bosonic variables [20] is to simulate bosonic quantum field theories [22], and wavelets are a very efficient choice of basis to discretize a quantum field theory for this purpose [23]. We explain that for any free 1+1-dimensional bosonic field theory, one can use suitably chosen biorthogonal wavelets to discretize the theory and use the corresponding wavelet decomposition to prepare its (approximate) ground state using the bosonic Gaussian entanglement renormalization circuit. The idea to use wavelets to discretize a field theory is quite natural, see for instance [23–25] for some recent discussions of discretizing bosonic field theories using wavelets. Our approach however fundamentally differs from these works in that we use *biorthogonal* wavelets (as is natural in the bosonic setting), which moreover are specifically designed to target the Hamiltonian of the field theory (rather than using off-the-shelf wavelets such as the Daubechies wavelets). We hope that our investigations can provide a potential starting point for the efficient simulation of interacting quantum field theories on quantum computers.

### 1.2 Organization of the paper

In Section 2 we give a brief review of biorthogonal filters and wavelet theory. In Section 3 we briefly review the formalism of quadratic bosonic Hamiltonians and Gaussian unitaries. We then explain the relation between entanglement renormalization and biorthogonal wavelet filters. In particular, in Section 3.1 we derive a relation the filters have to satisfy to disentangle the ground state of a given Hamiltonian. In Section 4 we explain how this gives rise to a circuit, and we state Section 4 which proves bounds on the accuracy of the approximation. Finally, in Section 5 we introduce continuous wavelet functions, and show that this gives a natural interpretation of entanglement renormalization in the corresponding quantum field theory. In the appendices we provide a review of the fermionic MERA/wavelet correspondence in Appendix A, an algorithm for constructing appropriate biorthogonal filters in Appendix B, an explanation of how to construct Gaussian circuits from filters in Appendix C and a precise statement and proof of Section 4 in Appendix D. Supporting code used to generate the numerical results in this work can be found at [26].

## 2 Perfect reconstruction and biorthogonal filters

Perfect reconstruction filters, or biorthogonal wavelet filters, are filters that decompose a signal into a high-frequency part and a low frequency part. This is reminiscent of the disentangling procedure of entanglement renormalization, and in this work we explain the precise connection. We first give a brief account of the theory of biorthogonal wavelet filters, see [14] for an

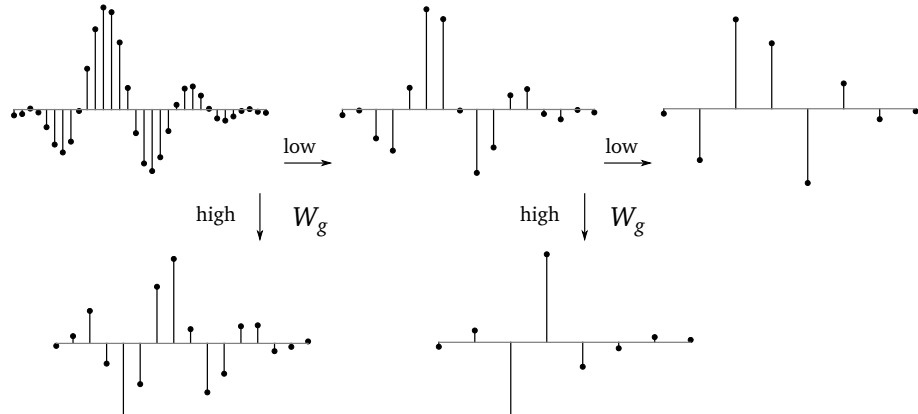

Figure 2: Iterating the wavelet decomposition $W_g$ resolves a signal into scales. Illustration for the Haar wavelet filters [19].

introduction. We consider a pair of real-valued sequences $g_s, h_s \in \ell^2(\mathbb{Z})$, called *scaling* or *low-pass filters*. Often they will be finite impulse response (FIR) filters, meaning that they have finite support. We demand that these filters satisfy the *perfect reconstruction condition* on their Fourier transforms

$$g_s(k)\overline{h_s(k)} + g_s(k+\pi)\overline{h_s(k+\pi)} = 2, \tag{1}$$

and define corresponding *wavelet* or *high-pass filters* by

$$g_w(k) = e^{-ik}\overline{h_s(k+\pi)} \quad \text{and} \quad h_w(k) = e^{-ik}\overline{g_s(k+\pi)}. \tag{2}$$

These filters can be used to separate a signal $\{f[n]\}_{n\in\mathbb{Z}}$ into a low-frequency and a high frequency component, and conversely to reconstruct the original signal from these components. For this, we let

$$f^{\text{low}}[n] = \sum_l g_s[l]f[2n+l],$$

$$f^{\text{high}}[n] = \sum_l g_w[l]f[2n+l],$$

and we define

$$W_g f = f^{\text{low}} \oplus f^{\text{high}}.$$

We similarly define $W_h$ using the filters $h_s$ and $h_w$ in place of $g_s$ and $g_w$, respectively. By applying $W_g$ again to $f^{\text{low}}$, the original signal is recursively resolved into scales, see Fig. 2. It follows from Eq. (1) that $f$ can be reconstructed from its decomposition $W_g f$ by applying the transposed operation $W_h^{\mathsf{T}}$, so $W_g^{-1} = W_h^{\mathsf{T}}$ and

$$f[n] = \sum_l h_s[n-2l]f^{\text{low}}[l] + h_w[n-2l]f^{\text{high}}[l].$$

The roles of $g$ and $h$ can be exchanged in this procedure.

## 3 Entanglement renormalization and filter design

We will consider translation invariant chains of harmonic oscillators $(q_n, p_n)$, with a Hamiltonian of the form

$$H = \frac{1}{2}\Big(\sum_{n\in\mathbb{Z}} p_n^2 + \sum_{n,m\in\mathbb{Z}} q_n V_{n-m} q_m\Big), \tag{3}$$

where $V_{nm} = V_{n-m}$ defines a positive definite symmetric matrix. The ground state of such a quadratic Hamiltonian is a Gaussian state, determined by the dispersion relation $\omega(k)$ of the Hamiltonian. We study Gaussian circuits that map an unentangled state to the entangled ground state of a translation invariant Hamiltonian (or conversely, disentangle the ground state to an unentangled state). We consider Gaussian maps defined by $\tilde{q}_n = \sum_m A_{nm} q_m$, $\tilde{p}_n = \sum_m B_{nm} p_m$. This preserves the canonical commutation relations if and only if the matrices $A$ and $B$ are such that $B = (A^\mathsf{T})^{-1}$. By a *Gaussian circuit* we will hence understand a sequence of Gaussian maps, each of which maps modes $(q_n, p_n)$ to a linear combination of itself and its direct neighbours. For details about quadratic bosonic Hamiltonians and Gaussian states see, for instance, [20, 27–29]. In Fourier space one simply maps a product state to the state with dispersion relation $\omega(k)$ by appropriately 'squeezing' each Fourier mode. However, this is a very *non-local* operation, whereas we are interested in a procedure that is local in real space. For more discussion of this point and variational algorithms to find Gaussian entanglement renormalization maps, see [21].

Since $W_g^{-1} = W_h^\mathsf{T}$ the map $W = W_g \oplus W_h$ defines a Gaussian map for any pair of biorthogonal wavelet filters $(g, h)$. This has the structure of a layer of entanglement renormalization, filtering out the high frequency modes. However, we need to choose the filters $g$ and $h$ such that $W$ actually disentangles the state, and the wavelet output is unentangled. If we normalize the dispersion relation such that $\omega(\pi) = 1$, then the condition for the wavelet output to be disentangled is that the Fourier transforms of the filters satisfy

$$g_w(k) = \omega(k) h_w(k). \tag{4}$$

Intuitively, what happens is that $W$ separates the bosonic modes in high frequency and low frequency modes, and Eq. (4) makes sure that the high frequency modes are not entangled to the low frequency modes in the ground state. We derive this condition below in Section 3.1. The scaling (low frequency) modes are again mapped to a Gaussian state, possibly with a different dispersion relation. We can now recursively apply the same construction to the scaling output, as in Fig. 1, now with the renormalized dispersion relation, given by

$$\omega(k) \mapsto \omega\left(\tfrac{k}{2}\right) \omega\left(\tfrac{k}{2} + \pi\right), \tag{5}$$

precomposed with a 'squeezing' normalization layer to ensure the normalization $\omega(\pi) = 1$ before we apply the wavelet decomposition. We will later see this procedure can be decomposed as a circuit, see Fig. 3.

While we motivated the procedure from the perspective of disentangling a given entangled state, the resulting circuit can also be used in the opposite direction, to prepare the ground state by applying the circuit to a product state, thus realizing the state as a bosonic MERA state. A paradigmatic example is the harmonic chain,

$$H = \frac{1}{2} \left( \sum_{n \in \mathbb{Z}} p_n^2 + m^2 q_n^2 + \frac{1}{4} (q_n - q_{n+1})^2 \right), \tag{6}$$

which has dispersion relation $\omega(k) = \sqrt{m^2 + \sin^2\left(\tfrac{k}{2}\right)}$. In particular, the *massless* harmonic chain is gapless and has dispersion relation $\omega(k) = |\sin\left(\tfrac{k}{2}\right)|$. For the latter, Eq. (5) amounts to $\omega(k) \mapsto |\sin\left(\tfrac{k}{4}\right) \cos\left(\tfrac{k}{4}\right)| = \frac{1}{2} \sin\left(\tfrac{k}{2}\right)$, so the dispersion relation is invariant under the renormalization step if we include the subsequent normalization. Hence the state on the scaling output of the entanglement renormalization will be the same after any number of layers. This implies that we can keep iterating the same entanglement renormalization layer with identical filters at each layer, giving a *scale-invariant* bosonic entanglement renormalization procedure for the massless harmonic chain.

In the massive case, the mass renormalizes as

$$m \mapsto 2\sqrt{m^2 + m^4}\,. \tag{7}$$

This is a *relevant* perturbation to the massless chain [21], and with increasing number of layers the dispersion relation becomes flat; correspondingly we can let the filters at the deeper layers approach orthogonal wavelet filters.

## 3.1 Derivation of filter condition

We will now derive Eq. (4). The ground state of the Hamiltonian in Eq. (3) is completely determined by its covariance matrix $\gamma = \gamma^q \oplus \gamma^p$, whose Fourier transform is given by

$$
\begin{aligned}
\gamma^q(k) &= \frac{1}{2\omega(k)}\,, \\
\gamma^p(k) &= \frac{\omega(k)}{2}\,.
\end{aligned}
\tag{8}
$$

The covariance matrix of an unentangled (uncorrelated) product state is $\frac{1}{2}\mathbb{1}$. Recall that any symplectic linear map $S$ on the set of modes $(q_n, p_n)$ defines a unitary map which maps Gaussian states to Gaussian states. Under a map of the form $A \oplus (A^{\mathsf{T}})^{-1}$ the covariance matrix transforms as

$$
\begin{aligned}
\gamma^q &\mapsto A\gamma^q A^{\mathsf{T}}\,, \\
\gamma^p &\mapsto (A^{\mathsf{T}})^{-1}\gamma^p A^{-1}\,.
\end{aligned}
$$

We first normalize such that $\omega(\pi) = 1$, which can be implemented by the symplectic (squeezing) map $(\sqrt{\omega(\pi)}\mathbb{1}) \oplus (1/\sqrt{\omega(\pi)}\mathbb{1})$. Suppose we have filters $(g, h)$ satisfying Eq. (4), then $W = W_g \oplus W_h$ disentangles the ground state. To see that this is indeed true, we compute the result of applying the wavelet decomposition map to the ground state covariance matrix $\gamma = \gamma^q \oplus \gamma^p$ given in terms of the dispersion relation by Eq. (8). For this, we remark that from $g_w(k) = \omega(k)h_w(k)$ it follows that $h_s(k) = \omega(k+\pi)g_s(k)$. Then,

$$
\begin{aligned}
\omega(k)h_w(k)f^{\text{high}}(2k) &= g_w(k)f^{\text{high}}(2k)\,, \\
\omega(k)h_s(k)f^{\text{low}}(2k) &= g_s(k)\omega^{(1)}(2k)f^{\text{low}}(2k)\,,
\end{aligned}
$$

where $\omega^{(1)}$ is the renormalized dispersion relation on the scaling output defined in Eq. (5) in the main text. This shows that $\omega(k)W_h^{\mathsf{T}} = W_g^{\mathsf{T}}(\omega^{(1)} \oplus \mathbb{1})$ and hence

$$
\begin{aligned}
W_h \gamma^p W_h^{\mathsf{T}} &= W_h W_g^{\mathsf{T}}(\gamma^{p,(1)} \oplus \frac{1}{2}\mathbb{1}) = \gamma^{p,(1)} \oplus \frac{1}{2}\mathbb{1}\,, \\
\gamma^{p,(1)}(k) &= \frac{1}{2}\omega^{(1)}(k)\,.
\end{aligned}
$$

Similarly, it holds that

$$
\begin{aligned}
W_g \gamma^q W_g^{\mathsf{T}} &= \gamma^{q,(1)} \oplus \frac{1}{2}\mathbb{1}\,, \\
\gamma^{q,(1)}(k) &= \frac{1}{2\omega^{(1)}(k)}\,.
\end{aligned}
$$

We thus see that $W$ has unentangled the high-frequency modes to a product state, and the low frequency modes are renormalized to have a new dispersion relation $\omega^{(1)}$ given by Eq. (5). The full entanglement renormalization circuit consists of repeated applications of such layers. To introduce some notation, we let $\omega^{(l)}$ be the dispersion relation after $l$ layers of

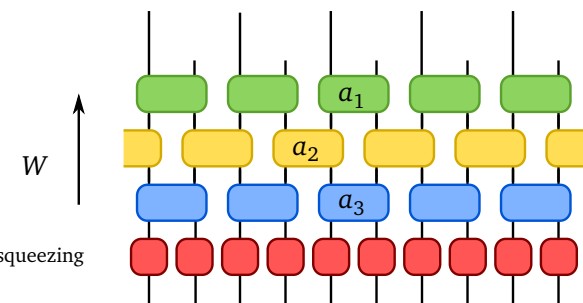

Figure 3: Decomposition of a single layer of the entanglement renormalization map $R^{(1)}$ as a circuit. The wavelet transform $W = W_g \oplus W_h$ is decomposed as a circuit with two-local gates $a_i$, and follows the bottom layer which squeezes by $\omega(\pi)^{\frac{1}{2}} \oplus \omega(\pi)^{-\frac{1}{2}}$ to normalize the dispersion relation. This figure can be interpreted both as a linear circuit implementing a symplectic transformation, and as its second quantization, which is a bosonic Gaussian circuit.

renormalization, recursively defined by (cf. Eq. (5), note that we first normalize the dispersion relation by a factor $\omega^{(l)}(\pi)$)

$$\omega^{(l+1)}(k) = \frac{\omega^{(l)}(\frac{k}{2})}{\omega^{(l)}(\pi)} \frac{\omega^{(l)}(\frac{k}{2} + \pi)}{\omega^{(l)}(\pi)}. \tag{9}$$

The normalization by $\omega^{(l)}(\pi)$ could also be absorbed in the filters, but we would like the filters to be such that $g(0) = h(0) = \sqrt{2}$, as is standard in the signal processing literature and convenient for the analysis. Then at the $l$-th layer we need filters $g^{(l)}$, $h^{(l)}$ satisfying $g_w^{(l)}(k) = \frac{\omega^{(l)}(k)}{\omega^{(l)}(\pi)} h_w^{(l)}(k)$ (cf. Eq. (4)), and we let

$$
\begin{aligned}
R_{g^{(l)}} &= W_{g^{(l)}} \sqrt{\omega^{(l)}(\pi)}, \\
R_{h^{(l)}} &= W_{h^{(l)}} \frac{1}{\sqrt{\omega^{(l)}(\pi)}}.
\end{aligned}
\tag{10}
$$

Finally, we define the $\mathcal{L}$-layer renormalization map as $R^{(\mathcal{L})} = R_g^{(\mathcal{L})} \oplus R_h^{(\mathcal{L})}$, where $R_a^{(\mathcal{L})} = (R_{a^{(\mathcal{L}-1)}} \oplus \mathbb{1}^{\oplus(\mathcal{L}-1)}) \circ \ldots \circ (R_{a^{(1)}} \oplus \mathbb{1}) \circ R_{a^{(0)}}$ for $a = g, h$. Then, $R^{(\mathcal{L})}$ maps the state with dispersion relation $\omega$ to a product state with covariance matrix $\frac{1}{2}\mathbb{1}$ on the $\mathcal{L}$ high frequency levels, and a state with dispersion relation $\omega^{(\mathcal{L})}$ on the remaining low frequency level.

## 4 Entanglement renormalization circuits

If $g$ and $h$ are FIR filters of size $2M$, we show in Appendix C that $W$ gives rise to a Gaussian circuit of depth $M$ that maps the low-frequency modes to the odd sublattice and the high-frequency modes to the even sublattice as shown in Fig. 3. This is exactly the structure of an entanglement renormalization circuit. The converse to this construction is also true: any Gaussian entanglement renormalization circuit as described above arises in this way.

When using a finite depth circuit, we may not be able to satisfy the relation in Eq. (4) exactly if $\omega(k)$ is not a ratio of trigonometric polynomials. In particular, this is the case for the harmonic chain. In this case we can still hope to *approximate* the dispersion relation, and correspondingly prepare a state that is close the true ground state. This raises two interesting questions. Firstly, the existence of filters that approximately satisfy Eq. (4) is not clear. In

Appendix B we describe an explicit procedure for constructing such filters. Secondly, one can wonder whether a good approximation of the dispersion relation at the level of a single layer will indeed give rise to a good approximation of the ground state. Fortunately, the structure of entanglement renormalization is remarkably robust to small errors [10, 30], and in [17] a robustness result for wavelet based fermionic entanglement renormalization was proven. The bosonic setting is somewhat different, as the Hilbert spaces are infinite dimensional. We will now discuss that when the family of filters has a well-defined 'continuum limit', we can nevertheless prove a rigorous approximation theorem.

We would like to bound the approximation error when using $\mathcal{L}$ layers of entanglement renormalization. Suppose we are given a family of filter pairs $(g^{(l)}, h^{(l)})$ for $l = 1, \ldots, \mathcal{L}$, where the $l$-th pair represents the $l$-th layer such that

$$\left| g_w^{(l)}(k) - \frac{\omega^{(l)}(k)}{\omega^{(l)}(\pi)} h_w^{(l)}(k) \right| \le \varepsilon, \qquad \forall l = 1, \ldots, \mathcal{L}, \tag{11}$$

so they approximately reproduce the dispersion relation at each layer (up to normalization). Moreover, we need these families of filters to give rise to a 'stable' wavelet decomposition, in the sense that many iterations of the decomposition maps yield a uniformly bounded map. This is a standard assumption in wavelet theory. If we are only interested in an approximation, and the theory flows to either a critical theory or a trivial theory we only need a small number of 'transition layers' and can pick fixed filters $(g^{(l)}, h^{(l)}) = (g, h)$ for large $l$. In Appendix D, we prove a general approximation theorem in this setting, which applies to an arbitrary quadratic Hamiltonian. We measure the error in the two-point functions $\langle p_i p_j \rangle$ and $\langle q_i q_j \rangle$ (or covariance matrix). In the particular case of the harmonic chain in Eq. (6), our result specializes as follows.

**Theorem** (Informal). *For the harmonic chain with mass $m$, the approximation error using the MERA state resulting from $\mathcal{L}$ layers of entanglement renormalization is bounded by*

$$|\langle p_i p_j \rangle_{\text{exact}} - \langle p_i p_j \rangle_{\text{MERA}}| \le \left( \mathcal{O}(2^{-\frac{\mathcal{L}}{2}}) + \mathcal{O}(\varepsilon \log \tfrac{1}{\varepsilon}) \right) \sqrt{m^2 + 1},$$

$$|\langle q_i q_j \rangle_{\text{exact}} - \langle q_i q_j \rangle_{\text{MERA}}| \le \left( \mathcal{O}(2^{-\frac{\mathcal{L}}{2}}) + \mathcal{O}(\varepsilon \log \tfrac{1}{\varepsilon}) \right) \frac{1}{m},$$

*the latter assuming $m > 0$. In the massless case, the latter bound is replaced by*

$$|\langle q_i q_j \rangle_{\text{exact}} - \langle q_i q_j \rangle_{\text{MERA}}| \le \left( O(2^{-\frac{\mathcal{L}}{2}}) + \mathcal{O}(\varepsilon \log \tfrac{1}{\varepsilon}) \right) \sqrt{|i - j|}.$$

In the massless case, there is an IR divergence and $\langle q_i q_j \rangle$ is only defined up to a constant, so we define $\langle q_i q_j \rangle$ by subtracting the divergence; see Eq. (47) in Appendix D for details.

The intuition behind the proof is that the contribution of the $\mathcal{L}$-th layer to the correlation function is bounded by $\mathcal{O}(2^{-\frac{\mathcal{L}}{2}})$, so we need $\mathcal{O}(\log \frac{1}{\delta})$ layers to get within error $\delta$ (even with perfect filters), while each layer contributes a factor of $\varepsilon$ to the error in the filter relation. Balancing these two contributions yields the desired bound. In Appendix B we provide a construction of filters $g, h$ satisfying Eq. (4) for the massless harmonic chain. This construction depends on two parameters $K$ and $L$, where $K$ controls the number of vanishing moments of the filters and $L$ controls the accuracy of the approximation of the dispersion relation. This corresponds to a circuit depth of $M = K = 2L$ for a single layer. In Fig. 4 we illustrate the approximation result by numerically computing correlation functions of the massless harmonic chain using these filters [26].

If we denote by $M$ the circuit depth of a single layer, then we find numerically that $\varepsilon$ is exponentially small as a function of $M$, whereas the other wavelet-dependent parameters we have suppressed above only grow polynomially. Hence, the total required depth of a single layer of entanglement renormalization for a desired error is polylogarithmic in $\frac{1}{\varepsilon}$. This shows that our entanglement renormalization circuits prepare the ground state very efficiently: a circuit of depth $\mathcal{O}(\text{polylog}(\frac{1}{\delta}))$ achieves an accuracy $\delta$ on the correlation functions.

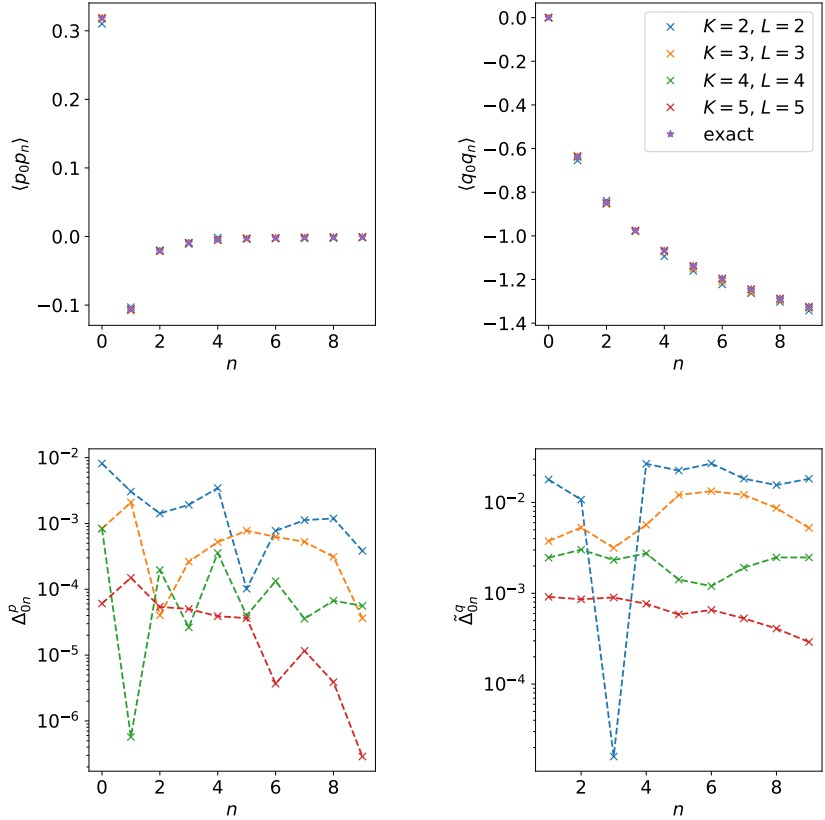

Figure 4: Approximation of correlation functions for the massless harmonic chain by the MERA. We used the filter construction of Appendix B and $\mathcal{L} = 20$ layers of renormalization. The former depends on parameters $K$ and $L$ which are explained in the main text. We show the correlation functions $\langle p_0 p_n \rangle$ and $\langle p_0 p_n \rangle$, as well as their approximation errors $\Delta_{0n}^p := |\langle p_0 p_n \rangle_{\text{exact}} - \langle p_0 p_n \rangle_{\text{MERA}}|$ and $\tilde{\Delta}_{0n}^q := |\langle q_0 q_n \rangle_{\text{exact}} - \langle q_0 q_n \rangle_{\text{MERA}}|$.

# 5  The continuum limit

The discrete wavelet transform has a natural continuum limit, in terms of continuous scaling and wavelet functions. This gives a way to interpret the continuous limit of the entanglement renormalization maps. In the following, we demonstrate that the scaling functions are a natural UV cut-off that is compatible with the entanglement renormalization circuits, and in the critical case we find that we can reproduce certain conformal data exactly from a single layer of renormalization. We consider the free boson, described by bosonic fields $\phi(x)$, $\pi(x)$ and Hamiltonian

$$H = \frac{1}{2} \int \mathrm{d}x \, \pi(x)^2 + m^2 \phi(x)^2 + \left( \partial \phi(x) \right)^2.$$

We are particularly interested in the massless case, which gives rise to a conformal field theory.

## 5.1  Scaling and wavelet functions

The continuum limit of the discrete biorthogonal wavelet transform is determined by the *scaling functions*. Given biorthogonal wavelet filters $g, h$ the associated scaling functions are defined in

Fourier space for $a = g, h$ by

$$\hat{\phi}^a(k) = \prod_{n=1}^{\infty} \frac{a_s(2^{-n}k)}{\sqrt{2}} \tag{12}$$

and the associated *wavelet functions* by

$$\hat{\psi}^a(k) = \frac{1}{\sqrt{2}} a_w\left(\frac{k}{2}\right)\hat{\phi}^a\left(\frac{k}{2}\right). \tag{13}$$

Both have compact support if the filters are finite, an example is shown in Fig. 5 . Moreover, we can define rescaled and shifted versions

$$\psi_{l,n}^a(x) = 2^{-\frac{l}{2}}\psi^a(2^{-l}x - n),$$
$$\phi_{l,n}^a(x) = 2^{-\frac{l}{2}}\phi^a(2^{-l}x - n).$$

It then follows that the sets $\{\psi_{l,n}^g\}_{l,n\in\mathbb{Z}}$ and $\{\psi_{l,n}^h\}_{l,n\in\mathbb{Z}}$ form a dual basis, in the sense that

$$\langle \psi_{l,n}^g, \psi_{l',n'}^h \rangle = \int \mathrm{d}x\, \psi_{l,n}^g(x)\psi_{l',n'}^h(x) = \delta_{l,l'}\delta_{n,n'}.$$

Moreover,

$$\langle \phi_{l,n}^g, \phi_{l,n'}^h \rangle = \int \mathrm{d}x\, \phi_{l,n}^g(x)\phi_{l,n'}^h(x) = \delta_{n,n'}.$$

If the filters are finite and the scaling functions are square-integrable functions (which is closely related to the discrete wavelet decomposition being sufficiently stable) the sets $\{\psi_{l,n}^g\}_{l,n\in\mathbb{Z}}$ and $\{\psi_{l,n}^h\}_{l,n\in\mathbb{Z}}$ form a Riesz basis of $L^2(\mathbb{R})$ [31]. This means that we can write any function $f \in L^2(\mathbb{R})$ as

$$f = \sum_{l,n}\langle \psi_{l,n}^g, f \rangle \psi_{l,n}^h = \sum_{l,n}\langle \psi_{l,n}^h, f \rangle \psi_{l,n}^g.$$

By construction of the scaling and wavelet functions, these are such that if

$$f = \sum_n s[n]\phi_{0,n}^g$$

then we can rewrite

$$f = \sum_{l=0}^{\mathcal{L}-1}\sum_n w[l,n]\psi_{l,n}^g + \sum_n \tilde{s}[n]\phi_{\mathcal{L},n}^g, \tag{14}$$

where we find the coefficients $w[l,n]$ and $\tilde{s}[n]$ precisely by applying the discrete wavelet transformation $W_h$ to the signal $s$. Moreover, if we let

$$f_l^h = \sum_n \langle \phi_{l,n}^g, f \rangle \phi_{l,n}^h, \tag{15}$$

$$f_l^g = \sum_n \langle \phi_{l,n}^h, f \rangle \phi_{l,n}^g, \tag{16}$$

then $f_l^h$ and $f_l^g$ converge in norm to $f$ as $l$ goes to minus infinity. See [14] for an introduction to scaling and wavelet functions and their properties.

## 5.2 Entanglement renormalization for the massless boson

We now suppose that the biorthogonal wavelet filters $g, h$ are related by the dispersion relation of the massless harmonic chain that was discussed in Section 3, that is,

$$g_w(k) = \left|\sin\left(\frac{k}{2}\right)\right| h_w(k). \tag{17}$$

We claim that, in this case, the scaling functions defined in Eq. (12) are related as

$$\hat{\phi}^g(k) = \frac{|k|}{2|\sin(\frac{k}{2})|} \hat{\phi}^h(k). \tag{18}$$

To verify this claim, we note that as a consequence of Eq. (17) and the relation in Eq. (2) we have $h_s(k) = |\cos(\frac{k}{2})| g_s(k)$. Next, from Eq. (12) it follows that $\hat{\phi}^h(k) = \gamma(k)\hat{\phi}^g(k)$, where

$$\gamma(k) = \prod_{n=1}^{\infty} |\cos(2^{-n-1}k)|.$$

This expression implies that $\gamma(k)$ has to satisfy $\gamma(k) = |\cos(\frac{k}{4})|\gamma(\frac{k}{2})$, and we can easily verify that $\gamma(k) = \frac{2|\sin(\frac{k}{2})|}{|k|}$, which has the right normalization $\gamma(0) = 1$. This proves Eq. (18), which in turn, using Eqs. (13) and (17), also implies that

$$\begin{aligned}
\hat{\psi}^g(k) &= \frac{1}{\sqrt{2}} g_w\left(\frac{k}{2}\right)\hat{\phi}^g\left(\frac{k}{2}\right) \\
&= \frac{1}{\sqrt{2}}\left|\sin\left(\frac{k}{4}\right)\right| h_w\left(\frac{k}{2}\right)\frac{|k|}{4|\sin(\frac{k}{4})|}\hat{\phi}^h\left(\frac{k}{2}\right) \\
&= \frac{|k|}{4}\hat{\psi}^h(k).
\end{aligned} \tag{19}$$

Equation (19) shows that wavelet functions are related precisely by the linear dispersion relation of the massless free boson.

We consider correlation functions of smeared fields $\phi(f) = \int dx\, f(x)\phi(x)$, $\pi(f) = \int dx\, f(x)\pi(x)$. First we consider the case where we have smeared fields $\phi(f)$ with $f$ of the form $f = \sum_n s[n]\phi^g_{l,n}$ and $\pi(\tilde{f})$ with $\tilde{f}$ of the form $\tilde{f} = \sum_n \tilde{s}[n]\phi^h_{l,n}$, then because the wavelet functions are precisely related by the correct dispersion relation, in order to compute correlation functions, it suffices to express the functions $f$ and $\tilde{f}$ in the wavelet bases $\{\psi^h_{l',n'}\}$ and $\{\psi^g_{l',n'}\}$. To see this it suffices to look at two-point functions, and suppose that we want to compute $\langle\pi_1(f_1)\pi(f_2)\rangle$, where $f_i = \sum_n s_i[n]\phi^h_{l,n}$. Then, if we rewrite $f_i = \sum_{l,n} w_i[l,n]\psi^h_{l,n}$ and we denote by $H$ the operator which is such that it acts as multiplication with $\frac{1}{4}|k|$ in the Fourier domain, so $H\psi^h = \psi^g$ and hence $2^l H\psi^h_{l,n} = \psi^g_{l,n}$, then

$$\begin{aligned}
\sum_{l,n} 2^{-l} w_1[l,n]w_2[l,n] &= \langle\sum_{l,n} w_1[l,n]\psi^h_{l,n}, \sum_{l',n'} 2^{-l'} w_2[l',n']\psi^g_{l',n'}\rangle \\
&= \langle\sum_{l,n} w_1[l,n]\psi^h_{l,n}, \sum_{l',n'} w_2[l',n']H\psi^h_{l',n'}\rangle \\
&= \langle f_1, H f_2\rangle,
\end{aligned}$$

which is indeed the correct correlation function. A similar computation holds for correlation functions involving the field $\phi$. However, by Eq. (14) the $w_i[l,n]$ are computed from $s_i[n]$ precisely by applying the discrete wavelet transform, and the factor of $2^{-l}$ derives from our

normalization of the dispersion relation (the 'squeezing layer' in Fig. 3). In other words, the correlation functions will be given precisely by applying the entanglement renormalization circuit to the operators $\sum_n s[n]q_n$ and $\sum_n \tilde{s}[n]p_n$. For general functions $f$, we may approximate them with scaling functions as in Eq. (15) and thus map

$$
\begin{aligned}
\phi(f) &\mapsto \sum_n \langle \phi_{l,n}^h, f \rangle q_n, \\
\pi(f) &\mapsto \sum_n \langle \phi_{l,n}^g, f \rangle p_n
\end{aligned}
\tag{20}
$$

(the inner product here is again the $L^2(\mathbb{R})$ inner product). The scale $l$ corresponds to a choice of UV cut-off. If $l$ is sufficiently small, then by Eq. (15) on can then compute correlation functions using the (discrete) entanglement renormalization circuit to good approximation. This approach is completely analogous to the fermionic construction described in detail in [19]. It yields a natural way to interpret the continuous limit of bosonic entanglement renormalization as quantum field theory.

As an application, we can consider the entanglement renormalization superoperator $\Phi$, which coarse-grains operators by conjugating with a single layer of the renormalization circuit. For critical lattice models, $\Phi$ has been proposed to approximately encode the conformal data of the continuum limit of the theory [9]. For instance, for a primary field in the conformal field theory with scaling dimension $\lambda$, there should be a local operator $O$ such that $\Phi(O) \approx 2^{-\lambda}O$. We will now verify that the entanglement renormalization superoperator reproduces *exactly* the scaling dimensions of the $\phi$ and $\pi$ fields in the massless case, as well as the scaling dimension of a number of descendants (equal to the number of vanishing moments of the wavelet filters), similar as for the fermionic wavelet MERA [16, 19]. This is seen by considering the operators $O_\phi(x) = \sum_n \phi^h(x-n)q_n$ and $O_\pi(x) = \sum_n \phi^g(x-n)p_n$ for any $x \in \mathbb{R}$, which are the discretizations of the operators $\phi(x)$ and $\pi(x)$. It can be easily seen that the entanglement renormalization superoperator maps these operators

$$
\begin{aligned}
O_\phi(x) &\mapsto \sum_{n,l} \sqrt{2} h_s[l] \phi^h(x-2n-l)q_n \\
&= \sum_n \phi^h\left(\frac{x}{2}-n\right)q_n = O_\phi\left(\frac{x}{2}\right),
\end{aligned}
$$

where we use that for the scaling function Eq. (12) it holds that [14]

$$
\frac{1}{\sqrt{2}} \phi^h\left(\frac{x}{2}\right) = \sum_n h_s[n]\phi^h(x-n).
\tag{21}
$$

Similarly one finds $O_\pi(x) \mapsto \frac{1}{2}O_\pi\left(\frac{x}{2}\right)$. This corresponds, as expected, to scaling dimensions 0 and 1. If the scaling function is differentiable, we see that by differentiating Eq. (21) we get that $\frac{1}{2\sqrt{2}}\partial_x \phi^h\left(\frac{x}{2}\right) = \sum_n h_s[n]\partial_x \phi^h(x-n)$, which leads similarly to a descendent field $O_{\phi^{(1)}} = \sum \partial_x \phi^h(x-n)q_n$ with the right scaling dimension. It turns out that if $\phi^h$ has $K$ vanishing moments (or equivalently, a factor $(1+e^{ik})^K$ in the scaling filter $h_s$ [14]), then there exists a vector $\phi^{h,l}[n]$ with $l = 1,\ldots,K$ and $n$ taking integer values on the support of the wavelet such that

$$
\frac{1}{2^l\sqrt{2}} \phi^{h,l}[m] = \sum_n h_s[n]\phi^{h,l}[2m-n],
$$

even if $\phi^h$ is not $l$ time differentiable (note that $\phi^{h,l}$ is only defined at integer values), see Theorem 7.1 in [32], and similarly for $\phi^{g,l}$. This shows that computing the eigenvalues of the

entanglement renormalization superoperator $\Phi$ will yield the eigenvalues of $K$ descendants of the $\phi$ and $\pi$ fields. At this point we observe that a wavelet filter leading to $K$ vanishing moments must have support at least $2K$, so one needs (as expected) a larger circuit depth to capture more descendent scaling dimensions. For example, in our explicit constructions in Appendix B the filter size is $2K + 4L$ where $L$ controls the accuracy of the approximation of the dispersion relation.

## 5.3 The massive bosonic field

The free massive boson with mass $m$ can be approached similarly. In that case, we suppose we have two families of filters $g^{(l)}$ and $h^{(l)}$, now with $l \in \mathbb{Z}$ and such that $\sqrt{(m^{(l)})^2 + 1}\, g_w^{(l)}(k) = \sqrt{(m^{(l)})^2 + \sin^2(\frac{k}{2})}\, h_w^{(l)}(k)$ where $m^{(0)} = m$ and $m^{(l)}$ is the mass after $l$ layers of renormalization, as defined by Eq. (7). If these filters are chosen in a way that they converge to a fixed *orthonormal* filter as $l$ goes to infinity, and to a fixed pair of biorthogonal filters as in the massless case for $l$ to $-\infty$, it makes sense to define a new type of scaling and wavelet functions which are different at each level $l$ as a generalization of of the scaling and wavelet functions:

$$
\begin{aligned}
\hat{\phi}_l^a(k) &= \prod_{j=1}^{\infty} \frac{a^{(l+j)}(2^{-j}k)}{\sqrt{2}}, \\
\hat{\psi}_l^a(k) &= \frac{1}{\sqrt{2}} a^{(l+1)}\Big(\frac{k}{2}\Big) \hat{\phi}_{l+1}^a\Big(\frac{k}{2}\Big),
\end{aligned}
\tag{22}
$$

for $a = g, h$ as a generalization of of the scaling and wavelet functions defined in Eq. (12) and Eq. (13). Again, the wavelet functions $\psi_{l,n}^a(x) = 2^{-\frac{l}{2}} \psi_l(2^{-l}x - n)$ for $a = g, h$ form a dual basis (provided they exist). The behaviour for $l \to \pm\infty$ is consistent with the fact that the mass term is a relevant perturbation of the conformal field theory and the theory flows from a critical massless boson to a trivial theory. As before, we can now discretize the theory using the scaling functions at some given scale and use the discrete circuit to compute correlation functions.

## 5.4 Other perspectives

The idea that wavelet theory should be a natural tool to discretize a field theory in order to perform renormalization has a long history [33]. As mentioned in the introduction, our approach differs from other works such as [23–25] which investigate the use of wavelets to discretize quantum field theories, in that we use biorthogonal wavelets, which moreover are specifically designed to target the Hamiltonian of the field theory. There is also a different approach to entanglement renormalization for quantum field theories, known as cMERA [34, 35]. This takes a different perspective by formulating a variational class of states directly in the continuum, rather than considering a discretization. In both cases, the correlation functions of the theory are accurately reproduced up to some cut-off. The precise relation between MERA and cMERA is not very well understood, for instance it is not clear that discretizing a cMERA state could yield a MERA. Intriguingly, cMERA is formally strongly reminiscent of the *continuous wavelet transform* (CWT). The continuous wavelet transform [14] can be defined for a much broader class of wavelet functions $\psi$, and if $\psi$ is a biorthogonal wavelet the CWT can be discretized to a discrete wavelet transform. Reformulating cMERA as the second quantization of a CWT would therefore give a clear relationship between MERA and cMERA for free bosonic systems. A starting point could be the cMERA in [36], which reproduces some scaling dimensions exactly. However, the CWT appears to break some of the symplectic properties of the discrete biorthogonal wavelet transform and it remains an open problem to make this connection more explicit. Finally, another reason why the field theory limit of entanglement renormalization is of interest is its tentative relation to holography in theories of quantum gravity, as conjectured

in [37]. The entanglement renormalization circuit can be thought of as mapping a system into one higher dimension by adding an additional 'scale' direction. An interpretation in terms of wavelets was proposed in [38] for fermions and extended to bosonic systems in [39].

# 6 Conclusion

In this work we have explained how Gaussian entanglement renormalization circuits can be naturally contructed from (and are in fact equivalent to) the second quantizations of biorthogonal wavelet transforms. There are a few technical aspects that would be interesting to study in more detail. First, one could carry out a fully rigorous analysis of the continuum limit discussed in Section 5, as in [19]. This poses some mathematical challenges. For example, if the system is not scale invariant then our notion of wavelet functions goes beyond the standard framework of wavelet theory, and one would have to identify suitable conditions on the filters that ensure that the wavelet and scaling functions as defined in Eq. (22) are well-behaved functions and that standard wavelet theory generalizes. Second, it would be desirable to identify conditions under which the procedure outlined in Appendix B is rigorously guaranteed to find good approximate solutions of Eq. (11). We note that even for Hilbert pair wavelets, which are relevant in the fermionic setting and which inspired our construction, this is not known and a subject of recent research in the signal processing community [40, 41].

Overall, we believe that this work, together with [16, 17] for the fermionic case, completes our conceptual understanding of Gaussian entanglement renormalization for free theories as the second quantization of wavelet decompositions. We hope that this offers a path towards constructing and analyzing entanglement renormalization circuits for *interacting* models. One clear direction is to apply perturbation theory in the wavelet basis. A similar approach has already been taken for cMERA in [42], where one can also do perturbation theory around a Gaussian cMERA. Another interesting direction is to investigate integrable models, where we know explicit solutions for the ground state, and try to formulate these in terms of wavelet modes. Finally, continuous wavelet transforms might also help illuminate the relation between MERA and cMERA as we discussed in Section 5.4.

# Acknowledgements

We thank Adrián Franco Rubio for inspiring discussions during the inception of this project.

**Funding information**   MW acknowledges funding by NWO Veni grant 680-47-459.

# A   Review of the fermionic wavelet-MERA correspondence

In this appendix we briefly review the construction of entanglement renormalization circuits for massless free fermions, as worked out in [16, 17, 19]. While not strictly needed to understand the results of the current work, which deals with free bosons, it is instructive to contrast the construction and state of the art with the fermionic setting. We closely follow the exposition in [17]. Let $a_n$ for $n \in \mathbb{Z}$ be fermionic operators, satisfying the anticommutation relations $\{a_n^\dagger, a_m\} = \delta_{n,m}$, $\{a_n, a_m\} = \{a_n^\dagger, a_m^\dagger\} = 0$. We work in the framework of Gaussian or free fermions. We consider Hamiltonians of the form

$$H = \sum_{n,m} h_{n,m} a_n^\dagger a_m,\qquad(23)$$

where $h$ is a Hermitian matrix. The ground state of such a Hamiltonian is given by

$$|\psi\rangle = \prod_i a^\dagger(f_i)|\Omega\rangle,$$

where $|\Omega\rangle$ is the Fock vacuum and $a^\dagger(f) := \sum_n f[n]a_n^\dagger$ and where the $f_i$ are all eigenvectors of $h$ with negative eigenvalue (so $\sum_m h_{n,m}f_i[m] = \lambda_i f_i[n]$ with $\lambda_i < 0$). In other words, precisely the negative energy modes are occupied in the ground state. The set of number-preserving Gaussian operations is given by all evolutions along Hamiltonians of the form in Eq. (23). Such transformations are the *fermionic second quantization* of unitaries acting on the single-particle space. That is, to every unitary operator $U$ acting on $\ell^2(\mathbb{Z})$ we associate the unitary which maps $a_n$ to $\tilde{a}_n = \sum_m u_{n,m}a_m$.

We restrict to the nearest neighbor hopping Hamiltonian

$$H = -\sum_{n\in\mathbb{Z}} a_n^\dagger a_{n+1} + a_{n+1}^\dagger a_n. \tag{24}$$

As opposed to the current work on bosonic models, so far no general wavelet construction for arbitrary free fermion models is known, but only for the Hamiltonian in Eq. (24) and its higher dimensional generalizations. To prepare the ground state of this Hamiltonian we separately consider the even and the odd sublattice and let $a_{1,n} = a_{2n}$ and $a_{2,n} = a_{2n+1}$, and we apply a phase gate, writing $b_{i,n} = (-1)^n a_{i,n}$. The Hamiltonian in Eq. (24) is then transformed to

$$H = -\sum_{n\in\mathbb{Z}} b_{1,n}^\dagger b_{2,n} - b_{2,n}^\dagger b_{1,n+1} + b_{2,n}^\dagger b_{1,n} - b_{1,n+1}^\dagger b_{2,n}. \tag{25}$$

This Hamiltonian can be written in Fourier space as

$$H = \int_{-\pi}^{\pi} \frac{dk}{2\pi} \begin{pmatrix} b_1(k) \\ b_2(k) \end{pmatrix}^\dagger \begin{pmatrix} 0 & e^{-ik}-1 \\ e^{ik}-1 & 0 \end{pmatrix} \begin{pmatrix} b_1(k) \\ b_2(k) \end{pmatrix}.$$

The ground state $|\psi\rangle$ is now given by filling the negative energy modes (the Fermi sea), which can be found by observing that for each $k$ the matrix

$$\begin{pmatrix} 0 & e^{-ik}-1 \\ e^{ik}-1 & 0 \end{pmatrix}$$

has a negative eigenvalue with eigenvector

$$\frac{1}{\sqrt{2}} \begin{pmatrix} 1 \\ -\operatorname{sgn}(k)ie^{i\frac{k}{2}} \end{pmatrix},$$

which means that the space of negative energy modes consists of all functions $f = (f_1, f_2)$ in $\ell^2(\mathbb{Z}) \otimes \mathbb{C}^2$ which are such that their Fourier transforms satisfy $f_2(k) = -\operatorname{sgn}(k)ie^{i\frac{k}{2}}f_1(k)$. That is, if $f$ is such a function, then $(b_1(f_1)^\dagger + b_2(f_2)^\dagger)|\psi\rangle = 0$. We now consider a pair of orthogonal wavelet filters (note that in contrast to the bosonic case, this is not a pair of biorthogonal filters, but two filters which are each orthogonal) $g_w$ and $h_w$, and we let $W_g$ and $W_h$ denote the corresponding wavelet decomposition maps, which are now orthogonal, i.e. $W_a^{-1} = W_a^\mathsf{T}$ for $a = g, h$. In particular, this means that we can apply the fermionic second quantization of $W_h$ to the $b_1$ fermions and $W_g$ to the $b_2$ fermions. If the filters are such that for $-\pi < k < \pi$

$$g_w(k) = -i\operatorname{sgn}(k)e^{i\frac{k}{2}}h_w(k), \tag{26}$$

Table 1: Comparison of the wavelet-MERA correspondence for fermions and bosons.

| | Fermions (at criticality) [16,17] | Bosons |
|---|---|---|
| Gaussian unitaries | $a_n \mapsto \sum_m U_{n,m} a_m$ for $U$ unitary | $q_n \mapsto A_{n,m} q_m$ and $p_n \mapsto A_{n,m} p_m$ with $A^{-1} = B^{\mathsf{T}}$ |
| Hamiltonian | $H = -\sum_n a_n^\dagger a_{n+1} + a_{n+1}^\dagger a_n$ | $H = \frac{1}{2}\left(\sum_{n\in\mathbb{Z}} p_n^2 + \sum_{n,m\in\mathbb{Z}} q_n V_{n-m} q_m\right)$ with dispersion relation $\omega(k)$ |
| Wavelet filters | $g, h$ orthogonal wavelet filters | $(g,h)$ pair of biorthogonal wavelet filters |
| Filter relation | $g_w(k) = -i\,\mathrm{sgn}(k)e^{i\frac{k}{2}}h_w(k)$ for $k \in (-\pi, \pi)$ | $g_w(k) = \omega(k)h_w(k)$ |
| Application of wavelet transform | $b_{1,n} = (-1)^n a_{2n}$, $b_{2,n} = (-1)^n a_{2n+1}$, and apply $W_h$ to the $b_1$ fermions and $W_g$ to the $b_2$ fermions | $A = W_g$, $B = W_h$ |
| Disentangling circuit | Apply wavelet decomposition, then $H$ on wavelet modes. | Apply squeezing to normalize dispersion relation, then apply the wavelet decomposition. |
| Continuum theory | Free Dirac fermion | Free bosonic scalar field (for $\omega(k) = |\sin(\frac{k}{2})|$) |
| Wavelet functions | $\hat{\psi}^g(k) = -i\,\mathrm{sgn}(k)\hat{\psi}^h(k)$ | $\hat{\psi}^g(k) = \frac{|k|}{4}\hat{\psi}^h(k)$ |

this will allow us to renormalize the ground state. To see this, consider any mode $f = (f_1, f_2)$ in the Fermi sea, so $f_2(k) = -\mathrm{sgn}(k)ie^{i\frac{k}{2}}f_1(k)$. When we apply the wavelet transforms $f_i$ is mapped to $(f_{i,w}, f_{i,s})$, a wavelet and a scaling component. Then one can show that from Eq. (26) it follows that $f_{1,w} = f_{2,w}$ and $f_{2,s}(k) = -\mathrm{sgn}(k)ie^{i\frac{k}{2}}f_{1,s}(k)$. We may now apply (the fermionic second quantization of) a Hadamard gate

$$H = \frac{1}{\sqrt{2}}\begin{pmatrix} 1 & 1 \\ 1 & -1 \end{pmatrix}$$

to the wavelet component, which then disentangles the wavelet component of the negative energy mode. This shows that if we take the the ground state $|\psi\rangle$ and we first apply the wavelet transforms $W_g$ and $W_h$ followed by $H$ on the wavelet output, we map to $|\psi\rangle$ to itself on the scaling output and the product state $\prod_n b_{1,n}^\dagger |\Omega\rangle$ (that is, the state in which all even sublattice modes $b_{1,n}$ are filled and all odd sublattice modes $b_{2n}$ are not filled) on the wavelet output. Thus we have implemented a layer of entanglement renormalization. One can write the fermionic second quantization of an orthogonal wavelet transform with a finite filter as a finite depth fermionic Gaussian circuit [18]. We may iteratively apply the renormalization to the scaling component to completely completely disentangle the ground state, and if we consider the circuit in the opposite direction then it maps layers of product states to the ground state of Eq. (25). One may think of this way of preparing the ground state as filling the Fermi sea layer by layer, now choosing a wavelet basis for the Fermi sea instead of the usual Fourier basis.

The relation Eq. (26) can not be satisfied exactly by a pair of finite filters, but it can be approximated. In [16] it was shown that a construction using Daubechies D4 filters already gives a good result, and in [17] a general method using known constructions from signal processing applications [40] was suggested, and it was also shown that a good approximation of Eq. (26) leads to a good approximation of the ground state (which inspired our Theorem 1). The Hamiltonian in Eq. (25) is in fact the Kogut-Susskind discretization of the Dirac fermion [43]. In [19] it has been worked out how the wavelet and scaling functions corresponding to the filters $g$ and $h$ have a natural interpretation in the quantum field theory, analogous to the discussion in Section 5. In Table 1 we give an overview of the analogies between the fermionic and bosonic case.

# B  Construction of filters

Next we will explain how to construct filter pairs that yield a good approximation of a given dispersion relation. Suppose we are given a dispersion relation $\omega(k)$. Let us assume that $\omega(k) = \omega(-k)$. We would like to construct a biorthogonal filter pair $(g, h)$ such that

$$g_w(k) \approx \omega(k) h_w(k) \tag{27}$$

or equivalently

$$h_s(k) \approx \omega(k + \pi) g_s(k). \tag{28}$$

We will describe a general approach to this problem inspired by the Daubechies wavelet construction, similar to the construction of Hilbert pair wavelets due to Selesnick [40] which were previously used in the construction of fermionic MERAs [17]. For this, we start with a rational approximation

$$\omega(k + \pi) \approx \frac{a(k)}{b(k)},$$

where $a$ and $b$ are real finite symmetric sequences on $[-L, L]$. The approximation only has to be accurate around $k = 0$. We will make the following ansatz for the Fourier transform of the scaling filters

$$
\begin{aligned}
g_s(k) &= b(k)(1 + e^{ik})^K f(k), \\
h_s(k) &= a(k)(1 + e^{ik})^K f(k),
\end{aligned}
\tag{29}
$$

where $f(k)$ is the Fourier transform of a real finite sequence $f[n]$ that still needs to be determined. The parameter $K$ determines the number of vanishing moments of the biorthogonal wavelets, just as in the Daubechies wavelet construction. By construction, $g_s(k)$ and $h_s(k)$ are small near $k = \pi$, and Eq. (28) is satisfied. In order for Eq. (29) to generate biorthogonal wavelet filters, they need to satisfy the condition in Eq. (1) which translates to

$$s(k)f(k)f(-k) + s(k + \pi)f(k + \pi)f(\pi - k) = 2,$$

where $s(k) = a(k)b(k)(2\cos(\frac{k}{2}))^{2K}$. One may try to solve this by letting $r(k) = f(k)f(-k)$. Then $r$ should be taken as a solution to the linear system

$$\sum_l s[2n - l]r[l] = \delta_0[n].$$

Now, if possible, we perform a spectral factorization $r(k) = f(k)f(-k)$. A necessary and sufficient condition for this is that $r(k) \geq 0$ for all $k$. Unfortunately, we do not know of a condition on $a$ and $b$ that guarantees this. The resulting filters $(g, h)$ will have support of size $2M$ where $M = K + 2L$. Finally, in the scale-invariant case, the stability condition that will be required in Theorem 1 can be checked explicitly for compactly supported filters by looking at the operators $P^g$ and $P^h$ defined by

$$(P^a f)(k) = |a(\frac{k}{2})|^2 f(\frac{k}{2}) + |a(\frac{k}{2} + \pi)|^2 f(\frac{k}{2} + \pi)$$

on the space of polynomials of degree at most $2M$ with zero mean. The filters yield square integrable scaling functions and uniformly bounded wavelet decomposition maps if and only if the eigenvalues of $P^g$ and $P^h$ are smaller in absolute value than 2 (see [44] or Theorem 4.2 in [45]).

For the massless harmonic chain, one particular choice for $a$ and $b$ is given by

$$
\begin{aligned}
a(k) &= \frac{1}{2}(e^{-iLk}d(k)^2 + e^{iLk}d(-k)^2), \\
b(k) &= d(k)d(-k),
\end{aligned}
\tag{30}
$$

where $d[n]$ is a maximally flat all-pass filter with delay $\frac{1}{4}$ of degree $L$ [40], so it has the property that $e^{-iLk}d(-k)/d(k) \approx e^{-i\frac{k}{2}}$ on $k \in (-\pi, \pi)$. In Fig. 6 we show the goodness of the approximation in Eqs. (27) and (28) as a function of $K$ and $L$. The resulting filters and wavelets for $K = 2$, $L = 4$ are shown in Fig. 5. We remark that the construction in Eq. (30) is not necessarily optimal. From numerical evidence in Fig. 6 it appears that the accuracy of the approximation improves exponentially with increasing support [26]. An interesting open problem is to rigorously prove the existence of approximate solutions to Eq. (27) with (exponentially) improving approximation accuracy as the filter size increases.

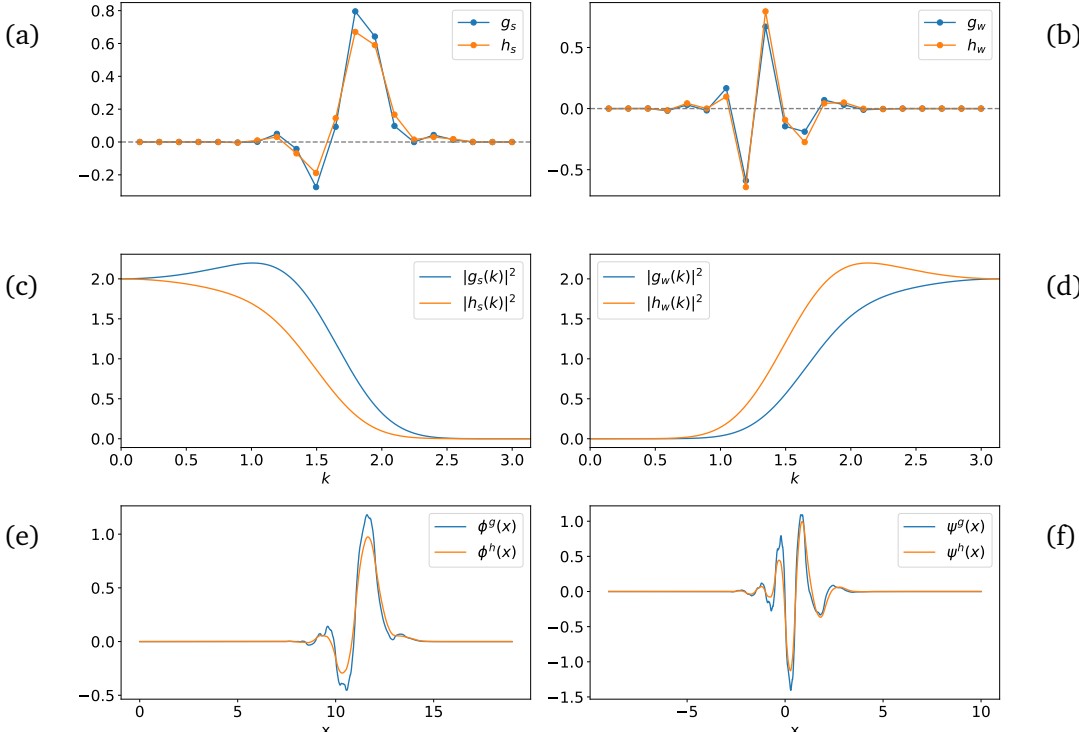

Figure 5: The results of using $K = 2$, $L = 4$ in the construction of Eq. (30): (a) scaling filters $g_s$ and $h_s$, (b) wavelet filters $g_w$ and $h_w$, (c) absolute value squared of the Fourier transforms of the scaling filters $|g_s(k)|^2$ and $|h_s(k)|^2$, (d) absolute value squared of the Fourier transforms of the wavelet filters $|g_w(k)|^2$ and $|h_w(k)|^2$, (e) scaling functions $\phi^g$ and $\phi^h$, and (f) wavelet functions $\psi^g$ and $\psi^h$.

## C  Construction of circuits from filters

We now discuss how to explicitly construct a Gaussian circuit from a given pair of biorthogonal wavelet filters, and show that any translation-invariant Gaussian circuit of the form of Fig. 3 always arises from such a filter pair.

Motivated by the fermionic setting it has been extensively discussed in [18] how to construct *unitary* local circuits from *orthogonal* wavelet filters. The construction for biorthogonal wavelet filters is very similar and the symmetric case has already been discussed in [18], but for completeness we provide it here. Given a pair of biorthogonal filters $(g, h)$ of support $2M$ we will construct a binary circuit of depth $M$ that implements the wavelet decomposition map. We will assume that $g$ and $h$ are supported on $[-M + 1, M]$, which we can always achieve by a

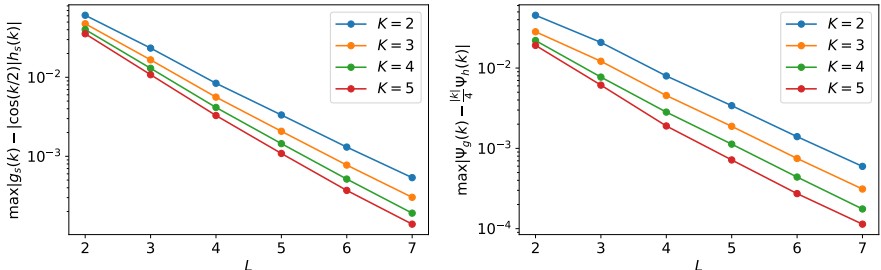

Figure 6: Approximation errors for $\varepsilon = \max_k |g_w(k) - |\sin(\frac{k}{2})|h_w(k)|$ and $\max_k |\psi^g(k) - \frac{|k|}{4}\psi^h(k)|$ for different values of $K$ and $L$ for a filter pair constructed using Eq. (30). For fixed $K$ the error appears to decrease exponentially in $L$.

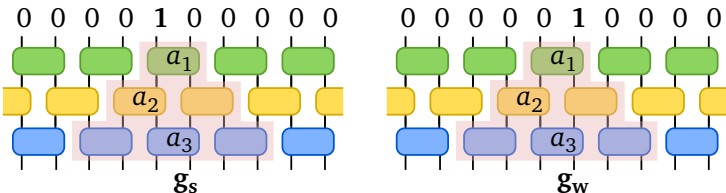

Figure 7: Illustration of Eq. (31), which gives the equations the $a_i$ have to satisfy in order for the circuit to implement $W_g$.

shift. By a binary circuit of depth $M$ we mean a sequence of maps $A_1, \ldots A_M$ on $\ell^2(\mathbb{Z})$ such that

$$A_i = \bigoplus_{n=\text{even}} (a_i)_{n,n+1},$$

for $i$ even, and similarly a sum over odd terms if $i$ is odd. Here $a_i$ is a two by two matrix. These maps will be such that $A = A_M \circ \ldots \circ A_1$ implements the wavelet reconstruction map in the sense that $Af = W_g^{\mathsf{T}}(f_{\text{odd}} \oplus f_{\text{even}})$ and $(A^{\mathsf{T}})^{-1}f = W_h^{\mathsf{T}}(f_{\text{odd}} \oplus f_{\text{even}})$ where $f_{\text{even}}[n] = f[2n]$ and $f_{\text{odd}}[n] = f[2n-1]$, as in Fig. 3. By shift invariance this is equivalent to

$$
\begin{aligned}
A\delta_1 &= g_s, \\
A\delta_2 &= g_w, \\
(A^{\mathsf{T}})^{-1}\delta_1 &= h_s, \\
(A^{\mathsf{T}})^{-1}\delta_2 &= h_w,
\end{aligned}
\tag{31}
$$

as illustrated for $g$ in Fig. 7. Now $A \oplus (A^{\mathsf{T}})^{-1}$ is a binary circuit, and its second quantization gives a Gaussian bosonic quantum circuit. It remains to construct the matrices $a_i$ given the filters $g$ and $h$. We will need the perfect reconstruction condition Eq. (1) which becomes

$$\sum_l g_s[2n+l]h_s[l] = \delta_0[n]$$

upon applying the inverse Fourier transform. In particular, the vectors $(g_s[-M+1], g_s[-M+2])^{\mathsf{T}}$ and $(h_s[M-1], h_s[M])^{\mathsf{T}}$ are orthogonal, and so are $(g_s[M-1], g_s[M])^{\mathsf{T}}$ and $(h_s[-M+1], h_s[-M+2])^{\mathsf{T}}$. Furthermore we will use that the wavelet filters are derived from the scaling filters as

$$
\begin{aligned}
g_w[n] &= (-1)^{(1-n)}h_s[1-n], \\
h_w[n] &= (-1)^{(1-n)}g_s[1-n],
\end{aligned}
\tag{32}
$$

which follows directly from Eq. (2). First suppose that $M = 1$. In that case we let

$$a_1 = \begin{pmatrix} g_s[0] & g_w[0] \\ g_s[1] & g_w[1] \end{pmatrix}.$$

Using the perfect reconstruction condition we may now check that

$$(a_1^\mathsf{T})^{-1} = \begin{pmatrix} h_s[0] & h_w[0] \\ h_s[1] & h_w[1] \end{pmatrix},$$

so this satisfies Eq. (31). For $M > 1$ we will construct the $A_i$ recursively. Let

$$g_M = \begin{pmatrix} g_s[M-1] & g_s[-M+2] \\ g_s[M] & g_s[-M+1] \end{pmatrix},$$

$$a_M = \frac{1}{\sqrt{\det(g_M)}} g_M,$$

then it is clear that $A_M^{-1}$ maps $g_s$ to a sequence $g_s^{(M-1)}$ on $[-M+2, M-1]$ and $A_M^\mathsf{T}$ maps $h_s$ to a sequence $h_s^{(M-1)}$ on $[-M+2, M-1]$ using the orthogonality properties derived from the perfect filter condition In the non-generic degenerate case that $\det(g_M) = 0$, the size of the support can only be decreased by 1 and an additional layer is needed. Moreover, since $A_M$ is invariant under shifts of 2, it is easy to see that $g_s^{(M-1)}$ and $h_s^{(M-1)}$ still satisfy the perfect reconstruction property. Finally, if we let $\alpha$ denote the map defined by $\alpha x[n] = (-1)^{(1-n)}x[1-n]$, then in order to see that $A_M^{-1}$ maps $g_w$ to the wavelet filter $g_w^{(M-1)}$ defined by $\alpha h_s^{(M-1)}$, it suffices to check that $A_M^{-1}\alpha = \alpha A_M^\mathsf{T}$ or equivalently $A_M^{-1} = \alpha A_M^\mathsf{T}\alpha$. This follows from the inversion formula for two by two matrices with determinant 1, i.e.,

$$\begin{pmatrix} a & b \\ c & d \end{pmatrix}^{-1} = \begin{pmatrix} d & -b \\ -c & a \end{pmatrix}.$$

Now we can recursively apply the same procedure to $(g^{(M-1)}, h^{(M-1)})$ to construct $A_{M-1}, \ldots, A_1$. We have now seen that we can construct a circuit from a filter pair.

Conversely, given a circuit of the form $A = A_M \circ \ldots \circ A_1$ as described above, *define* filters $g$ and $h$ by Eq. (31). We can then check that these filters form perfect reconstruction filters, in the sense that $W_h^\mathsf{T} = W_g^{-1}$. If we assume $\det(a_i) = 1$ for $i = 1, \ldots, M$, the wavelet and scaling filters are related as in Eq. (32).

## D  Approximation theorem

In this appendix we state and prove a general approximation result for translation-invariant quadratic Hamiltonians of the form Eq. (3). We obtain the theorem in the main text by specializing to the harmonic chain (as explained at the very end of this appendix). Our proof strategy is inspired by the techniques in [17], with the technical complications that the wavelet transforms are not unitary and are allowed to vary layer by layer.

If $g_s, h_s \in \ell^2(\mathbb{Z})$ are a pair of scaling filters that satisfy the perfect reconstruction condition in Eq. (1) of the main text, then we can define corresponding wavelet filters $g_w, h_w \in \ell^2(\mathbb{Z})$ and single-layer decomposition maps $W_g, W_h : \ell^2(\mathbb{Z}) \to \ell^2(\mathbb{Z}) \oplus \ell^2(\mathbb{Z})$ such that $W_h^T W_g = W_g^T W_h = \mathbb{1}$.

Now suppose that we are given a sequence of filters $g_s^{(l)}, h_s^{(l)}$ as above. Here, $l \in \mathbb{N}$ for convenience of notation. In practice, one is usually interested in a finite number of layers; in

this case we may choose the sequence of filters to eventually become constant. For $a = g, h$ and $\mathcal{L} \in \mathbb{N}$, we define the $\mathcal{L}$-layer decomposition maps

$$W_a^{(\mathcal{L})} \colon \ell^2(\mathbb{Z}) \to \ell^2(\mathbb{Z})^{\otimes(1+\mathcal{L})},$$
$$W_a^{(\mathcal{L})} := \left(W_{a^{(\mathcal{L}-1)}} \oplus \mathbb{1}^{\oplus(\mathcal{L}-1)}\right) \circ \ldots \circ \left(W_{a^{(1)}} \oplus \mathbb{1}\right) \circ W_{a^{(0)}},$$

and write $W^{(\mathcal{L})} = W_h^{(\mathcal{L})} \oplus W_g^{(\mathcal{L})}$. We assume that the family is *stable* in the sense that the corresponding (generalized) scaling functions $\phi_l^a$ defined in Eq. (22) exist, are square integrable, and bounded in $L^\infty$-norm. We can also define the wavelet decomposition maps starting at layer $\mathcal{L}' \geq 0$, that is,

$$W_a^{(\mathcal{L}',\mathcal{L})} \colon \ell^2(\mathbb{Z}) \to \ell^2(\mathbb{Z})^{\otimes(1+\mathcal{L}-\mathcal{L}')},$$
$$W_a^{(\mathcal{L}',\mathcal{L})} := \left(W_{a^{(\mathcal{L})}} \oplus \mathbb{1}^{\oplus(\mathcal{L}-\mathcal{L}'-1)}\right) \circ \ldots \circ \left(W_{a^{(\mathcal{L}'+1)}} \oplus \mathbb{1}\right) \circ W_{a^{(\mathcal{L}')}}.$$

For $\mathcal{L}' = 0$ we recover $W_a^{(\mathcal{L})}$ as defined earlier. We assume that the wavelet decomposition maps are bounded. Finally, we shall assume that the filters have finite support. Then the same is true for the scaling functions. In the case that the filters are independent of $l$, the above notion of stability is equivalent to the familiar notion from wavelet theory. For finitely supported filters there exists an easy criterion to determine this, see [31].

For the entanglement renormalization circuit, we also insert a squeezing operation between each wavelet decomposition layer, defining $R_{a^{(l)}}$, $R_a^{(\mathcal{L})}$ and $R^{\mathcal{L}}$ for $a = g, h$ as in Eq. (10). Our approximation to the covariance matrix is then given by

$$\gamma_{\text{MERA}}^{q,(\mathcal{L})} = \frac{1}{2} R_h^{(\mathcal{L}),\mathsf{T}} R_h^{(\mathcal{L})},$$
$$\gamma_{\text{MERA}}^{p,(\mathcal{L})} = \frac{1}{2} R_g^{(\mathcal{L}),\mathsf{T}} R_g^{(\mathcal{L})}. \tag{33}$$

Suppose the filter pairs $g^{(l)}, h^{(l)}$ approximately satisfy the renormalized dispersion relation at each level as in Eq. (11). That is,

$$\left| g_w^{(l)}(k) - \frac{\omega^{(l)}(k)}{\omega^{(l)}(\pi)} h_w^{(l)}(k) \right| = |g_w^{(l)}(k) - \tilde{g}_w^{(l)}(k)| \leq \varepsilon, \tag{34}$$

where we have introduced the filter

$$\tilde{g}_w^{(l)}(k) := \frac{\omega^{(l)}(k)}{\omega^{(l)}(\pi)} h_w^{(l)}(k). \tag{35}$$

This filter, together with $\tilde{h}_w^{(l)}(k) := \omega^{(l)}(\pi)/\omega^{(l)}(k) \times g_w^{(l)}(k)$, forms a pair of biorthogonal wavelet filters, with corresponding scaling filters $\tilde{g}_s^{(l)}, \tilde{h}_s^{(l)}$ that satisfy Eq. (1). However, these filters are almost never finitely supported. By construction, $\tilde{g}^{(l)}, h^{(l)}$ satisfy Eq. (4) exactly.

We now state our approximation theorem for general dispersion relations. We measure the approximation error in terms of quantities

$$\Delta_{nm}^p := |\gamma_{nm}^p - (\gamma_{\text{MERA}}^{p,(\mathcal{L})})_{nm}|,$$
$$\Delta_{nm}^q := |\gamma_{nm}^q - (\gamma_{\text{MERA}}^{q,(\mathcal{L})})_{nm}|. \tag{36}$$

If $\omega(0) = 0$, then it is also interesting to regulate the covariance matrix $\gamma^q$ as

$$\tilde{\gamma}_{nm}^q := \gamma_{nm}^q - \gamma_{nn}^q \tag{37}$$

and consider

$$\tilde{\Delta}_{nm}^q := |\tilde{\gamma}_{nm}^q - (\tilde{\gamma}_{\text{MERA}}^{q,(\mathcal{L})})_{nm}|. \tag{38}$$

**Theorem 1.** *Consider a translation-invariant Hamiltonian of the form of Eq. (3), with dispersion relation $\omega(k)$ such that $\omega^{(l)}(\pi) \leq 1$ and $\omega^{(l)}(k) \leq \Omega$ for $l = 1, \ldots, \mathcal{L}$ for some $\Omega \geq 1$. Suppose we have a sequence of filters such that Eq. (34) holds for $\varepsilon \leq 1$, with finite support of size at most $M$ and scaling functions that are uniformly bounded by $\|\phi_l^a\|_\infty \leq B$ for $a = g, h$ and $l = 1, \ldots, \mathcal{L}$. Assume moreover that the wavelet decomposition maps are uniformly bounded by $\|W_a^{(l',l)}\| \leq D$ for all $a = g, h, \tilde{g}$ and $1 \leq l \leq l' \leq \mathcal{L}$, where $D \geq 1$. Then the approximation error of the covariance matrices can be bounded as follows:*

$$\Delta_{nm}^p \leq D^2 \Big( C 2^{-\frac{\mathcal{L}}{2}} + 3\varepsilon D \log_2 \frac{C}{\varepsilon} \Big),$$

$$\Delta_{nm}^q \leq 2D^2 \Big( C 2^{-\frac{\mathcal{L}}{2}} + 3\varepsilon D \log_2 \frac{C}{\varepsilon} \Big) \|\gamma^q \delta_0\|,$$

$$\tilde{\Delta}_{nm}^q \leq 2D^2 \Big( C 2^{-\frac{\mathcal{L}}{2}} + 3\varepsilon D \log_2 \frac{C}{\varepsilon} \Big) \|\gamma^q (\delta_n - \delta_m)\|,$$

*where $C := 4B^2 M^{\frac{3}{2}} \Omega$.*

To interpret the error bounds, we note that

$$\|\gamma^q \delta_0\|^2 = \int_{-\pi}^{\pi} \frac{dk}{\omega(k)^2}, \tag{39}$$

$$\|\gamma^q (\delta_n - \delta_m)\|^2 = \int_{-\pi}^{\pi} dk \, \frac{\sin^2(\frac{1}{2}(n-m)k)}{\omega(k)^2}. \tag{40}$$

As mentioned earlier, our proof strategy follows [17] with two technical complications: the wavelet transforms are not unitary and are allowed to vary layer by layer.

We will first bound the error that arises from only taking a finite number of layers. Let $p_s^{(\mathcal{L})}$ denote the projection onto the first tensor factor of $\ell^2(\mathbb{Z})^{\otimes(\mathcal{L}+1)}$ and $p_w^{(\mathcal{L})} = \mathbb{1} - p_s^{(\mathcal{L})}$ the projection onto the remaining tensor factors. Thus, $p_s^{(\mathcal{L})} W_a^{(\mathcal{L})} f$ is the scaling component of the decomposed signal and $p_w^{(\mathcal{L})} W_a^{(\mathcal{L})}$ its wavelet component. The following lemma confirms the intuition that, for finitely supported signals, lower-frequency wavelet modes contribute less.

**Lemma 1.** *Suppose we have sequence of filters as above, with finite support of size at most $M$ and scaling functions that are uniformly bounded by $\|\phi_l^a\|_\infty \leq B$ for $a = g, h$ and $l = 1, \ldots, \mathcal{L}$. Then,*

$$\|p_s^{(\mathcal{L})} W_a^{(\mathcal{L})} \delta_n\| \leq 2^{-\frac{\mathcal{L}-1}{2}} B^2 M^{\frac{3}{2}}, \tag{41}$$

*where $\delta_n$ is the unit signal concentrated at $n$.*

*Proof.* Let $b$ denote the filters dual to $a$ (i.e., $b = h$ if $a = g$, and vice versa). We note that $p_s^{(\mathcal{L})} W_a^{(\mathcal{L})} \delta_n[m] = \langle \phi_{0,n}^b, \phi_{\mathcal{L},m}^a \rangle$, where $\phi_{\mathcal{L},m}^a(x) := 2^{-\mathcal{L}/2} \phi_{\mathcal{L}}^a(2^{-\mathcal{L}} x - m)$ are the translated and shifted scaling functions. This follows from the fact that $\langle \phi_{0,n}^b, \phi_{0,m}^a \rangle = \delta_{nm}$ and by applying inductively the fact that by definition of the scaling functions $\phi_{l+1,m}^a = \sum_n a_s^{(l+1)}(2m-n)\phi_{l,n}^a$.

Now we can proceed as in the proof of Lemma 1 in [17] and estimate

$$
\begin{aligned}
\|p_s^{(\mathcal{L})} W_a^{(\mathcal{L})} \delta_n\|^2 &= \sum_m \Big| \int_{-\infty}^{\infty} dx\, \phi_0^b(x-n) 2^{-\frac{\mathcal{L}}{2}} \phi_{\mathcal{L}}^a(2^{-\mathcal{L}}x - m) \Big|^2 \\
&= \sum_m \Big| \int_{x_0+n}^{x_0+n+M-1} dx\, \phi_0^b(x-n) 2^{-\frac{\mathcal{L}}{2}} \phi_{\mathcal{L}}^a(2^{-\mathcal{L}}x - m) \Big|^2 \\
&\leq \sum_m \|\phi_0^b\|^2 \int_{x_0+n}^{x_0+n+M-1} dx\, |2^{-\frac{\mathcal{L}}{2}} \phi_{\mathcal{L}}^a(2^{-\mathcal{L}}x - m)|^2 \\
&= 2^{-\mathcal{L}} \|\phi_0^b\|^2 \sum_m \int_{x_0+n}^{x_0+n+M-1} dx\, |\phi_{\mathcal{L}}^a(2^{-\mathcal{L}}x - m)|^2 \\
&\leq 2^{-\mathcal{L}+1} M^2 \|\phi_0^b\|^2 \|\phi_{\mathcal{L}}^a\|_\infty^2,
\end{aligned}
$$

where in the second line we use that $\phi^b$ is compactly supported on $[x_0, x_0 + M - 1]$ for some $x_0$, in the third inequality we use Cauchy-Schwarz, and for the final inquality we use that at most $2M$ terms in the sum have nonzero overlap. Finally we may estimate $\|\phi_0^b\|^2 \leq MB^2$ and $\|\phi_{\mathcal{L}}^a\|_\infty^2 \leq B^2$, which yields Eq. (41). $\qquad\square$

The following lemma bounds the approximation error for $\mathcal{L}$ layers as a function of an intermediate layer $\mathcal{L}'$ that will later be chosen appropriately.

**Lemma 2.** *Suppose we have a sequence of filters such that Eq. (34) holds, with finite support of size at most $M$ and scaling functions that are uniformly bounded by $\|\phi_l^a\|_\infty \leq B$ for $a = g, h$ and $l = 1, \dots, \mathcal{L}$. Assume moreover that the wavelet decomposition maps are uniformly bounded by $\|W_a^{(l',l)}\| \leq D$ for all $a = g, h, \tilde{g}$ and $1 \leq l \leq l' \leq \mathcal{L}$, where $D \geq 1$. Finally, let $\mathcal{L}' \in \{1, \dots, \mathcal{L}\}$. Then we have the following bounds:*

(i) *For all $f \in \ell^2(\mathbb{Z})$ and $n \in \mathbb{N}$,*

$$
|\langle \delta_n | \gamma^q - \gamma_{\text{MERA}}^{q,(\mathcal{L})} | f \rangle| \leq 2D^2 \left( \varepsilon \mathcal{L}' D + 2^{-\frac{\mathcal{L}'-1}{2}} B^2 M^{\frac{3}{2}} \max\{2\|\gamma^{p,(\mathcal{L})}\|, 1\} \right) \|\gamma^q f\|. \tag{42}
$$

(ii) *Assuming $\omega^{(l)}(\pi) \leq 1$ for all $l = 0, \dots, \mathcal{L} - 1$, we have the following bound for all $n \in \mathbb{N}$:*

$$
\|(\gamma^p - \gamma_{\text{MERA}}^{p,(\mathcal{L})}) \delta_n\| \leq D^2 \left( \varepsilon \mathcal{L}' D + 2^{-\frac{\mathcal{L}'-1}{2}} B^2 M^{\frac{3}{2}} \max\{2\|\gamma^{p,(\mathcal{L}')}\|, 1\} \right). \tag{43}
$$

*Here, we recall that $\gamma^q(k) = \frac{1}{2\omega(k)}$ and $\gamma^{p,(l)}(k) = \frac{1}{2}\omega^{(l)}(k)$.*

To interpret these bounds, we note that $\|\gamma^{p,(\mathcal{L})}\| = \max_k \frac{\omega^{(\mathcal{L})}(k)}{2}$, which is typically $\mathcal{O}(1)$. As a remark, for the critical harmonic chain we that $\omega^{(l)}(\pi) = \frac{1}{2}$, in which case it is not hard to see that the scaling of Eq. (43) can be improved to $2^{-\frac{3}{2}\mathcal{L}'}$.

*Proof of Lemma 2.* (i) To prove Eq. (42), we first observe that by definition of $\tilde{g}$ it holds that

$$
R_{h^{(l)}} \omega^{(l)}(k) = (\omega^{(l+1)} \oplus \mathbb{1}) R_{\tilde{g}^{(l)}}
$$

and hence

$$
R_h^{(\mathcal{L})} \gamma^p = (\gamma^{p,(\mathcal{L})} \oplus \frac{1}{2}\mathbb{1}) R_{\tilde{g}}^{(\mathcal{L})}, \tag{44}
$$

where $\gamma^{p,(\mathcal{L})}(k) = \frac{1}{2}\omega^{(\mathcal{L})}(k)$ denotes the covariance matrix defined using the renormalized dispersion relation. We use this, together with the fact that $4\gamma^p\gamma^q = \mathbb{1}$ on the domain of $\gamma^q$ to write

$$
\begin{aligned}
\gamma^q - \gamma^{q,(\mathcal{L})}_{\text{MERA}} f &= (I - \gamma^{q,(\mathcal{L})}_{\text{MERA}} 4\gamma^p)\gamma^q f \\
&= (W_h^{(\mathcal{L}'),\mathsf{T}} W_g^{(\mathcal{L}')} - R_h^{(\mathcal{L}),\mathsf{T}} R_h^{(\mathcal{L})} 2\gamma^p)\gamma^q f \\
&= (W_h^{(\mathcal{L}'),\mathsf{T}} W_g^{(\mathcal{L}')} - R_h^{(\mathcal{L}),\mathsf{T}} (2\gamma^{p,(\mathcal{L})} \oplus \mathbb{1}) R_{\tilde{g}}^{(\mathcal{L})})\gamma^q f \\
&= (W_h^{(\mathcal{L}'),\mathsf{T}} W_g^{(\mathcal{L}')} - W_h^{(\mathcal{L}),\mathsf{T}} (2\gamma^{p,(\mathcal{L})} \oplus \mathbb{1}) W_{\tilde{g}}^{(\mathcal{L})})\gamma^q f
\end{aligned}
$$

for $f$ in the domain of $\gamma^q$. Thus,

$$
\begin{aligned}
|\langle \delta_n | \gamma^q - \gamma^{q,(\mathcal{L})}_{\text{MERA}} | f \rangle| \leq\ & |\langle \delta_n | W_h^{(\mathcal{L}'),\mathsf{T}} p_s^{(\mathcal{L}')} W_g^{(\mathcal{L}')} | \gamma^q f \rangle| \\
& + |\langle \delta_n | W_h^{(\mathcal{L}'),\mathsf{T}} p_w^{(\mathcal{L}')} \big( W_g^{(\mathcal{L}')} - W_{\tilde{g}}^{(\mathcal{L}')} \big) | \gamma^q f \rangle| \\
& + |\langle \delta_n | (W_h^{(\mathcal{L}),\mathsf{T}} (2\gamma^{p,(\mathcal{L})} \oplus \mathbb{1}) W_{\tilde{g}}^{(\mathcal{L})} - W_h^{(\mathcal{L}'),\mathsf{T}} p_w^{(\mathcal{L}')} W_{\tilde{g}}^{(\mathcal{L}')}) | \gamma^q f \rangle|.
\end{aligned}
\tag{45}
$$

We will bound the three terms separately, starting with the second term. By our assumption on the filters (Eq. (34)), $\|W_{g^{(l)}} - W_{\tilde{g}^{(l)}}\| \leq 2\varepsilon$. Hence, using a telescoping sum,

$$
\|W_g^{(\mathcal{L}')} - W_{\tilde{g}}^{(\mathcal{L}')}\| \leq \sum_{l=0}^{\mathcal{L}'-1} \|W_g^{(l+1,\mathcal{L}')}\| \|W_{g^{(l)}} - W_{\tilde{g}^{(l)}}\| \|W_{\tilde{g}}^{(l)}\| \leq 2\varepsilon \mathcal{L}' D^2,
\tag{46}
$$

so we obtain the estimate

$$
|\langle \delta_n | W_h^{(\mathcal{L}'),\mathsf{T}} p_w^{(\mathcal{L}')} \big( W_g^{(\mathcal{L}')} - W_{\tilde{g}}^{(\mathcal{L}')} \big) | \gamma^q f \rangle| \leq \|W_h^{(\mathcal{L}')}\| \|W_g^{(\mathcal{L}')} - W_{\tilde{g}}^{(\mathcal{L}')}\| \|\gamma^q f\| \leq 2\varepsilon \mathcal{L}' D^3 \|\gamma^q f\|.
$$

The first term in Eq. (45) can be bounded directly using Lemma 1,

$$
|\langle \delta_n | W_h^{(\mathcal{L}'),\mathsf{T}} p_s^{(\mathcal{L}')} W_g^{(\mathcal{L}')} | \gamma^q f \rangle| \leq \|p_s^{(\mathcal{L}')} W_h^{(\mathcal{L}')} \delta_n\| \|W_g^{(\mathcal{L}')} \gamma^q f\| \leq 2^{-\frac{\mathcal{L}'-1}{2}} B^2 M^{\frac{3}{2}} D \|\gamma^q f\|,
$$

and the third term may be similarly bounded as

$$
\begin{aligned}
& |\langle \delta_n | (W_h^{(\mathcal{L}),\mathsf{T}} (2\gamma^{p,(\mathcal{L})} \oplus \mathbb{1}) W_{\tilde{g}}^{(\mathcal{L})} - W_h^{(\mathcal{L}'),\mathsf{T}} p_w^{(\mathcal{L}')} W_{\tilde{g}}^{(\mathcal{L}')}) | \gamma^q f \rangle| \\
&= |\langle \delta_n | (W_h^{(\mathcal{L}'),\mathsf{T}} (W_h^{(\mathcal{L}',\mathcal{L}),\mathsf{T}} \oplus \mathbb{1}^{\oplus \mathcal{L}'}) (2\gamma^{p,(\mathcal{L})} \oplus \mathbb{1}^{\oplus(\mathcal{L}-\mathcal{L}')} \oplus 0^{\oplus \mathcal{L}'}) W_{\tilde{g}}^{(\mathcal{L})} | \gamma^q f \rangle| \\
&= |\langle \delta_n | (W_h^{(\mathcal{L}'),\mathsf{T}} p_s^{(\mathcal{L}')} (W_h^{(\mathcal{L}',\mathcal{L}),\mathsf{T}} \oplus 0^{\oplus \mathcal{L}'}) (2\gamma^{p,(\mathcal{L})} \oplus \mathbb{1}^{\oplus \mathcal{L}}) W_{\tilde{g}}^{(\mathcal{L})} | \gamma^q f \rangle| \\
&\leq \|p_s^{(\mathcal{L}')} W_h^{(\mathcal{L}')} \delta_n\| \|W_h^{(\mathcal{L}',\mathcal{L})}\| \|2\gamma^{p,(\mathcal{L})} \oplus \mathbb{1}^{\oplus \mathcal{L}}\| \|W_{\tilde{g}}^{(\mathcal{L})}\| \|\gamma^q f\| \\
&\leq 2^{-\frac{\mathcal{L}'-1}{2}} B^2 M^{\frac{3}{2}} D^2 \max\{2\|\gamma^{p,(\mathcal{L})}\|, 1\} \|\gamma^q f\|.
\end{aligned}
$$

By combining the three estimates we obtain Eq. (42).

(ii) To prove Eq. (43) we use Eqs. (33) and (44) to write

$$
\begin{aligned}
\gamma^p - \gamma^{p,(\mathcal{L})}_{\text{MERA}} &= \gamma^p - \frac{1}{2} R_g^{(\mathcal{L}),\mathsf{T}} R_g^{(\mathcal{L})} = \gamma^p R_h^{(\mathcal{L}'),\mathsf{T}} R_g^{(\mathcal{L}')} - \frac{1}{2} R_g^{(\mathcal{L}),\mathsf{T}} R_g^{(\mathcal{L})} \\
&= R_{\tilde{g}}^{(\mathcal{L}'),\mathsf{T}} (\gamma^{p,(\mathcal{L}')} \oplus \frac{1}{2} \mathbb{1}) R_g^{(\mathcal{L}')} - \frac{1}{2} R_g^{(\mathcal{L}),\mathsf{T}} R_g^{(\mathcal{L})}.
\end{aligned}
$$

Therefore,

$$
\begin{aligned}
\|(\gamma^p - \gamma^{p,(\mathcal{L})}_{\text{MERA}}) \delta_n\| \leq\ & \|R_{\tilde{g}}^{(\mathcal{L}'),\mathsf{T}} \gamma^{p,(\mathcal{L}')} p_s^{(\mathcal{L}')} R_g^{(\mathcal{L}')} \delta_n\| \\
& + \frac{1}{2} \|(R_{\tilde{g}}^{(\mathcal{L}'),\mathsf{T}} - R_g^{(\mathcal{L}'),\mathsf{T}}) p_w^{(\mathcal{L}')} R_g^{(\mathcal{L}')} \delta_n\| \\
& + \frac{1}{2} \|(R_g^{(\mathcal{L}'),\mathsf{T}} p_w^{(\mathcal{L}')} R_g^{(\mathcal{L}')} - R_g^{(\mathcal{L}),\mathsf{T}} R_g^{(\mathcal{L})}) \delta_n\|.
\end{aligned}
$$

As before we bound the three terms separately, starting with the second term. Since $\omega^{(l)}(\pi) \leq 1$, we may estimate $\|R_g^{(\mathcal{L}')}\| \leq \|W_g^{(\mathcal{L}')}\| \leq D$ and $\|R_{\tilde{g}}^{(\mathcal{L}')} - R_g^{(\mathcal{L}')}\| \leq 2\varepsilon \mathcal{L}' D^2$ by a telescoping sum as in Eq. (46). Thus:

$$\frac{1}{2}\|(R_{\tilde{g}}^{(\mathcal{L}'),\mathsf{T}} - R_g^{(\mathcal{L}'),\mathsf{T}})p_w^{(\mathcal{L}')}R_g^{(\mathcal{L}')}\delta_n\| \leq \frac{1}{2}\|(R_{\tilde{g}}^{(\mathcal{L}'),\mathsf{T}} - R_g^{(\mathcal{L}'),\mathsf{T}})\|\|R_g^{(\mathcal{L}')}\| \leq \varepsilon \mathcal{L}' D^3.$$

For the remaining terms, we note that $\omega^{(l)}(\pi) \leq 1$ also implies that $\|p_s^{(\mathcal{L}')}R_g^{(\mathcal{L}')}\delta_n\| \leq \|p_s^{(\mathcal{L}')}W_g^{(\mathcal{L}')}\delta_n\| \leq 2^{-\frac{\mathcal{L}'-1}{2}}B^2 M^{\frac{3}{2}}$ using Lemma 1. We can thus bound the first term by

$$\|R_{\tilde{g}}^{(\mathcal{L}'),\mathsf{T}}\gamma^{p,(\mathcal{L}')}p_s^{(\mathcal{L}')}R_g^{(\mathcal{L}')}\delta_n\| \leq \|R_{\tilde{g}}^{(\mathcal{L}'),\mathsf{T}}\|\|\gamma^{p,(\mathcal{L}')}\|\|p_s^{(\mathcal{L}')}R_g^{(\mathcal{L}')}\delta_n\| \leq 2^{-\frac{\mathcal{L}'-1}{2}}B^2 M^{\frac{3}{2}}D\|\gamma^{p,(\mathcal{L}')}\|,$$

and similarly the third term, where we find

$$\begin{aligned}\frac{1}{2}\|(R_g^{(\mathcal{L}),\mathsf{T}}R_g^{(\mathcal{L})} - R_g^{(\mathcal{L}'),\mathsf{T}}p_w^{(\mathcal{L}')}R_g^{(\mathcal{L}')})\delta_n\| &= \frac{1}{2}\|R_g^{(\mathcal{L}),\mathsf{T}}R_g^{(\mathcal{L}',\mathcal{L})}p_s^{(\mathcal{L}')}R_g^{(\mathcal{L}')}\delta_n\| \\ &\leq \frac{1}{2}\|R_g^{(\mathcal{L}),\mathsf{T}}\|\|R_g^{(\mathcal{L}',\mathcal{L})}\|\|p_s^{(\mathcal{L}')}R_g^{(\mathcal{L}')}\delta_n\| \\ &\leq \frac{1}{2}2^{-\frac{\mathcal{L}'-1}{2}}B^2 M^{\frac{3}{2}}D^2.\end{aligned}$$

By combining the three estimates we obtain Eq. (43). $\qquad\square$

We finally prove our general approximation theorem.

*Proof of Theorem 1.* Choosing $\mathcal{L}' = \min\{\lfloor 2\log_2 \frac{C}{\varepsilon}\rfloor, \mathcal{L}\}$, we see that

$$\begin{aligned}\varepsilon\mathcal{L}'D + 2^{-\frac{\mathcal{L}'-1}{2}}B^2 M^{\frac{3}{2}}\max\{2\|\gamma^{p,(\mathcal{L}')}\|, 1\} &\leq \varepsilon\mathcal{L}'D + 2^{-\frac{\mathcal{L}'-1}{2}}B^2 M^{\frac{3}{2}}\Omega \\ &\leq 2\varepsilon D\log_2\frac{C}{\varepsilon} + \max\{C2^{-\frac{\mathcal{L}}{2}}, \varepsilon\} \\ &\leq 3\varepsilon D\log_2\frac{C}{\varepsilon} + C2^{-\frac{\mathcal{L}}{2}},\end{aligned}$$

where we have used that $\frac{C}{\varepsilon} \geq 2$. Now the result follows from Eq. (43) and Eq. (42) in Lemma 2, choosing $f = \delta_m$ or $f = \delta_m - \delta_n$ in the latter (and using $\|\gamma^q\delta_m\| = \|\gamma^q\delta_0\|$). $\qquad\square$

Finally, we claim that the theorem in the main text is just the specialization of Theorem 1 to the harmonic chain with mass $m$. The correlation functions are related to the covariance matrices as follows:

$$\begin{aligned}\langle p_i p_j\rangle &= \gamma_{ij}^p, \\ \langle q_i q_j\rangle &= \gamma_{ij}^q,\end{aligned}$$

the latter assuming $m > 0$. If $m = 0$ then the latter has a divergence, so we instead define

$$\langle q_i q_j\rangle := \gamma_{ij}^q - \gamma_{ii}^q = \tilde{\gamma}_{ij}^q, \tag{47}$$

where $\tilde{\gamma}^q$ is the regulated covariance matrix defined in Eq. (37). Accordingly, we would like to bound the quantities $\Delta_{ij}^p$ as well as $\Delta_{ij}^q$ (in the massive case) or $\tilde{\Delta}_{ij}^q$ (in the massless case), which are defined in Eqs. (36) and (38). This is exactly achieved by Theorem 1. We first normalize the dispersion relation of the harmonic chain $\omega(k)$ by a factor $\sqrt{m^2 + 1}$ to $\omega_{\text{norm}}$, so that $\omega_{\text{norm}}(\pi) = 1$. There $\omega_{\text{norm}}^{(l)}(k) \leq 1$ and we may apply Theorem 1 with $\Omega = 1$. We write $\gamma$ for the original covariance matrix of the harmonic chain and $\gamma_{\text{norm}}$ for the covariance matrix where

the dispersion relation has been normalized, that is, $\gamma_{\text{norm}}^p = \frac{1}{\sqrt{m^2+1}}\gamma^p$ and $\gamma_{\text{norm}}^q = \sqrt{m^2+1}\gamma^q$. Then,

$$\frac{1}{m^2+1}\|\gamma_{\text{norm}}^q \delta_0\|^2 = \|\gamma^q \delta_0\|^2 = \int_{-\pi}^{\pi} \frac{dk}{\omega(k)^2} = \int_{-\pi}^{\pi} \frac{dk}{m^2 + \sin^2\left(\frac{k}{2}\right)} \leq \frac{2\pi}{m^2},$$

so applying Theorem 1 using the covariance matrix $\gamma_{\text{norm}}$ and restoring the factor $\sqrt{m^2+1}$ yields the results for $\Delta_{ij}^p$ and $\Delta_{ij}^q$. In the massless case we can use Eq. (40) and estimate

$$\begin{aligned}
\|\gamma^q(\delta_i - \delta_j)\|^2 &= \int_{-\pi}^{\pi} dk \, \frac{\sin^2(\frac{|i-j|k}{2})}{\sin^2(\frac{k}{2})} \\
&\leq 2\left( \int_0^{\frac{2}{|i-j|}} dk \, \frac{\pi^2 |i-j|^2}{4} + \int_{\frac{2}{|i-j|}}^{\pi} dk \, \frac{\pi^2}{k^2} \right) \\
&\leq 2\pi^2 |i-j|,
\end{aligned}$$

since $|\sin(\frac{k}{2})| \geq \frac{|k|}{\pi}$ and $|\sin(\frac{nk}{2})| \leq \min\{\frac{n|k|}{2}, 1\}$ on the interval $(-\pi, \pi)$, yielding the estimate for $\tilde{\Delta}_{ij}^q$.

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
