# Peer review of "Bosonic entanglement renormalization circuits from wavelet theory"

_SciPost Physics, doi:SciPost Phys. 10, 143 (2021)_

## Round 1 · Referee Report · Anonymous (Referee 4) · 2021-5-9

Strengths

The authors
1- illustrate a new approach for discretizing bosonic fields using biorthogonal wavelets
2- provide an in-depth mathematical illustration for translation invariant chains of harmonic oscillators
3- introduce a thought-threw estimation for the approximation error introduced by performing the entanglement renormalisation
4- provide an extensive appendix supporting their approximation theorem

Weaknesses

1- The manuscript is of theoretical nature and could be imporved by a more extensive numerical study benchmarking the approach e.g. against exactly solvable ground state preparations or alternative techniques
2- There is already a rich literature on the connection between wavelet theory and entanglement renormalisation. In this context, the results might not necessarily come unexpected.

Report

The authors address all the points I raised in my previous report. Especially the restructured introduction improves the readability of the manuscript and offers the reader a better introduction into the topic including some of the relevant literature. I appreciate as well, the expanded sections 5.2 and 5.3 on the continuous wavelet functions adding more details to the discussion on the continuum limit. With all the changes made, I believe the manuscript is a round, solid and sound work.

The authors raise a valid point that the fermionic case is yet restricted to a critical model of hopping fermions while the here presented approach, in general, works for arbitrary free bosonic models. With this in mind, I came to appreciate the work more and can see that this work might be suited for a publication in SciPost Physics. However, compared to other works, I am still left with the feeling that the paper is indeed interesting, though extremely technical, and the main results somehow expected (as the referee report of Luca states as well).

After I went through the literature again, I have to admit that, while I am active in the field of Tensor Networks and numerical renormalisation, I might not have the profound overview on the field of wavelet theory to confidently judge whether the scientific importance is high enough for a publication in SciPost Physics or if the manuscript is better suited for SciPost Physics Core. Thus, I would leave this decision to the other referees and the editor in charge.

In any case, I would still have a few minor, cosmetic remarks which are listed below.

Requested changes

1- Regarding point 8 of the last report (the claims in the introduction of Sec. 5), the authors replied that these claims are discussed in Sec. 5.2 and 5.3 . This is correct and completely satisfying (especially with the now expanded sections 5.2 and 5.3). However, for the sake of clarity, I would recommend changing the wording from "It turns out the scaling functions ..." to something like "In Sec. 5.2 and 5.3, we discuss how the scaling functions..." or "In the following, we demonstrate that the scaling functions...". Just to tell the reader that this claim is yet to be discussed in the manuscript.

2- In the abstract, the authors write "We give a general algorithm [...] and prove an approximation result that...". The phrasing of "proving an approximation result" sounds strange to me. I would suggest "provide an approximation theory", "provide an approximation result" or "introduce an approximation result/theory".

3- In Fig. 4, the authors use the parameters L and K, which are not introduced in the context of the figure but rather in the appendix. I would suggest to (i) either include a short description of the parameters in the main text near Fig. 4 or (ii) to move the figure to the appendix and referring to the appendix in the main text.

Further, the figure shows the computed correlations for different modes of the harmonic oscillator and the exact values. Since they seem to overlap nicely, it could be more expressive to plot here the actual deviation of the computations with respect to the exact results, i.e. |<p_0 p_n>_{K,L} - <p_0 p_n>_{exact}| rather then |<p_0 p_n>_{K,L}|.

---

## Round 1 · Author Response

We would like to thank the editor and the reviewers for their thoughtful consideration of our work.
We believe that their feedback and suggestions have allowed us to improve the quality and exposition of our work.
We reply in detail to each of the reviewer reports with an Author Reply.

---

## Round 1 · List of Changes

We have
- Expanded the introduction
- Added an outline of the organization of the paper
- Added numerics of correlation functions
- Significantly expanded sections 5.2 and 5.3 on continuous wavelet functions
- Added an appendix reviewing the fermionic MERA/wavelet correspondence and a comparison with the bosonic case
- Improved the presentation throughout the manuscript.
See the replies to the individual reviewers for more details on these changes.

---

## Round 2 · Author Response

We would like to thank the reviewers for their positive response, and Reviewer 1 for the suggestions for improvement.

---

## Round 2 · List of Changes

We have implemented all improvements suggested by Reviewer 1.
In addition, we have made the following changes:
- Some cosmetic changes in Appendix D and below the informal approximation theorem in Section 4, where we now cross-reference the precise regularization used for the reader's convenience.
- We have made available and refer to the code that constructs the appropriate wavelet filters and generates the figures in our work, so that the numerical results are easily reproducible (Ref. 26).

---

## Editorial Decision

published